## OPEN

# Evolution of stickleback spines through independent *cis*-regulatory changes at *HOXDB*

Julia I. Wucherpfennig [1], Timothy R. Howes [2], Jessica N. Au [1], Eric H. Au[3],
Garrett A. Roberts Kingman [1], Shannon D. Brady[1], Amy L. Herbert[1], Thomas E. Reimchen[4],
Michael A. Bell[5], Craig B. Lowe [3], Anne C. Dalziel[6] and David M. Kingsley [1,7] ✉

Understanding the mechanisms leading to new traits or additional features in organisms is a fundamental goal of evolutionary biology. We show that *HOXDB* regulatory changes have been used repeatedly in different fish genera to alter the length and number of the prominent dorsal spines used to classify stickleback species. In *Gasterosteus aculeatus* (typically 'three-spine sticklebacks'), a variant *HOXDB* allele is genetically linked to shortening an existing spine and adding an additional spine. In *Apeltes quadracus* (typically 'four-spine sticklebacks'), a variant *HOXDB* allele is associated with lengthening a spine and adding an additional spine in natural populations. The variant alleles alter the same non-coding enhancer region in the *HOXDB* locus but do so by diverse mechanisms, including single-nucleotide polymorphisms, deletions and transposable element insertions. The independent regulatory changes are linked to anterior expansion or contraction of *HOXDB* expression. We propose that associated changes in spine lengths and numbers are partial identity transformations in a repeating skeletal series that forms major defensive structures in fish. Our findings support the long-standing hypothesis that natural *Hox* gene variation underlies key patterning changes in wild populations and illustrate how different mutational mechanisms affecting the same region may produce opposite gene expression changes with similar phenotypic outcomes.

The origins of diverse vertebrate body plans have fascinated comparative anatomists and evolutionary biologists for centuries[1,2]. Although studies over the last 40 years have identified many cellular pathways required for body axis formation and development using induced mutations in model organisms, it is challenging to identify specific changes in genes and regulatory regions that underlie the diversity of body forms and traits in wild species[3,4].

*Hox* genes were one of the first classes of major developmental genes to be identified and analysed in comparative studies across animals. They were initially discovered by linked clusters of mutations in *Drosophila* that could transform particular body segments into others[5]. Molecular studies revealed that *Hox* loci consist of clustered homeodomain transcription factor genes, whose expression patterns along the anterior–posterior body axis were correlated with their physical positions along the chromosome[6-10].

In an early review of genetic work on homeotic loci, Lewis[5] hypothesized that regulatory mutations in *Hox* genes might underlie classic anterior–posterior patterning differences between species, such as four-winged versus two-winged insects. Although subsequent studies showed that *Hox* expression patterns are conserved between two-winged fruit flies and four-winged butterflies[11,12], the important role of *Hox* genes in controlling body patterning has led to speculation that mutations in these genes underlie key morphological differences in nature[10,13]. Variation in *Hox* cluster number and structure across taxa support this idea, and intriguing correlations can be drawn between morphological differences and *Hox* expression changes[10,14,15]. On the other hand, much of the diversification

and expansion of *Hox* clusters occurred before well-known morphological changes among animal phyla[16]. Furthermore, many laboratory mutations in *Hox* genes lead to reduced viability or fertility, and prominent evolutionary biologists[17,18] and critics of evolutionary biology[19] have suggested that natural mutations in *Hox* genes would lead to 'hopeless monsters' rather than adaptive changes in wild species. Natural differences in leg trichomes and abdominal pigmentation have been linked to genetic variation in *Hox* loci in insects, with regulatory mutations providing a possible mechanism for bypassing the broader deleterious consequences seen with many laboratory mutations[20,21]. However, few detailed examples exist for the long-postulated idea that genetic changes in *Hox* loci may also be the basis for major changes in skeletal structures along the anterior–posterior body axis of wild vertebrates[15,22].

Almost a third of extant vertebrate species fall in the large and diverse Acanthomorpha group of spiny-rayed fish[23], many of which show dramatic changes in the size or number of axial skeletal structures. A key evolutionary innovation of this group is the development of stiff, unsegmented bony spines anterior to the median dorsal and anal fins. These dorsal spines can be raised to protect against predators or lowered to facilitate swimming[24]. The number, length and morphology of bony spines differ substantially among species[25]. Recent studies have begun to reveal how spines form and grow within the median fin fold of developing fish[26-28]. However, little is known about the molecular changes that underlie the diverse spine patterns seen in different species.

Sticklebacks form a diverse clade of fish within Acanthomorpha. Multiple genera of sticklebacks live in northern marine and

[1]Department of Developmental Biology, Stanford University School of Medicine, Stanford, CA, USA. [2]Department of Chemical and Systems Biology, Stanford University School of Medicine, Stanford, CA, USA. [3]Department of Molecular Genetics and Microbiology, Duke University School of Medicine, Durham, NC, USA. [4]Department of Biology, University of Victoria, Victoria, British Columbia, Canada. [5]University of California Museum of Paleontology, University of California, Berkeley, CA, USA. [6]Department of Biology, Saint Mary's University, Halifax, Nova Scotia, Canada. [7]Howard Hughes Medical Institute, Stanford University School of Medicine, Stanford, CA, USA. ✉e-mail: kingsley@stanford.edu

freshwater environments and diverged over 16 million years ago[29–31]. The most well studied of these species, *Gasterosteus aculeatus*, also known as the three-spine stickleback, colonized new freshwater postglacial habitats from the oceans after widespread melting of glaciers approximately 12,000 years ago[32]. In new freshwater environments containing different food sources and predators, *Gasterosteus* populations evolved substantial differences in craniofacial structures, vertebrae and the number of defensive bony plates and spines along the anterior–posterior body axis[32]. Many recently evolved populations show major reductions of structures, including armour plate loss, pelvic hind fin loss, spine length reduction and reduced body pigmentation[28,33–35]. However, recently derived populations can also evolve increases in size or number of structures, including increased body size, increased number of teeth, increased spine length and increased dorsal spine number[27,36–38].

In this study, we used genetic and genomic approaches in two different stickleback genera to study the molecular mechanisms involved in spine patterning changes in natural populations. Our studies provide new evidence to support the long-standing hypothesis that mutations in the *cis*-regulatory regions of *Hox* genes underlie the evolution of new skeletal patterns along the anterior–posterior body axis of wild vertebrate species.

## Results

**Quantitative trait locus mapping of spine number and length in *Gasterosteus*.** To study the genetics of spine number in *Gasterosteus aculeatus*, we generated an F2 cross by crossing a wild-caught female freshwater stickleback from Boulton Lake, British Columbia, Canada and a wild-caught male marine stickleback from Bodega Bay, California, USA. The marine fish had the three dorsal spines typically seen in *Gasterosteus*. The freshwater fish had 2 dorsal spines (Extended Data Fig. 1), as is true for 80% of the *Gasterosteus* found in Boulton; the other 20% of fish have 3 dorsal spines[39]. We intercrossed F1 males and females with 3 dorsal spines and raised 590 F2 offspring (Fig. 1a). Most F2 individuals had 3 dorsal spines ($n = 563$), but 6 had 2 spines and 21 had 4 spines (Extended Data Fig. 1). We numbered spines from anterior to posterior, with the posterior-most spine immediately in front of the dorsal fin called dorsal spine last (DSL). Therefore, a four-spine *Gasterosteus* has dorsal spine 1 (DS1), dorsal spine 2 (DS2), dorsal spine 3 (DS3) and DSL, which we refer to as high-spine. A typical three-spine *Gasterosteus* has DS1, DS2 and DSL, which we refer to as low-spine in this study (Extended Data Fig. 2a).

To examine the genetic basis of morphological phenotypes along the anterior–posterior body axis, we genotyped 340 F2 individuals with a custom single-nucleotide polymorphism (SNP) array (Methods)[40] and phenotyped fish for the number and length of dorsal spines, number of flat bony plates that form in the dorsal midline or at the base of spines, known as pterygiophores, and the number of abdominal, caudal and total vertebrae (Fig. 1b and

Extended Data Fig. 2). There were not enough two-spine fish for mapping the two- versus three-spine trait. When mapping three-versus four-spine as a categorical trait, we detected a significant quantitative trait locus (QTL) on the distal end of chromosome 6 (percentage variance explained: 5.8%). The same chromosome region showed a significant QTL for DS2 length (percentage variance explained: 8.2%). The allele linked to both increased spine number and decreased DS2 length was inherited from the freshwater Boulton parent. None of the vertebral traits mapped to the distal end of chromosome 6 (Extended Data Fig. 2), suggesting that the effect of this chromosome region was specific to patterning dorsal spines but not to axial patterning as a whole. Previous studies of other populations have identified other loci controlling vertebral number[41,42].

**HOXD11B is in the candidate interval and expressed in spines.** The distal end of chromosome 6 contains the *HOXDB* locus in *Gasterosteus*. While not annotated in the original reference genome (*gasAcu*1 (ref. [43])), previous studies of *Hox* clusters across multiple species suggest that the *Gasterosteus* locus includes three genes (*HOXD11B*, *HOXD9B* and *HOXD4B*) and one microRNA (*miR-10d*)[44]. *Hox* genes are known to be expressed in the neural tube and somites as the body axis forms[45,46]. To investigate *HOXDB* gene expression in sticklebacks, we used in situ hybridization during embryonic axis formation (stage 19/20)[47]. *HOXD4B* was expressed in the hindbrain, neural tube and anterior-most somites; *HOXD9B* was expressed more posteriorly in the somites and neural tube, and *HOXD11B* was expressed in the most posterior somites and tailbud (Extended Data Fig. 3), which is consistent with similar colinear patterns in other organisms[10,15].

Dorsal spines form weeks after early embryonic patterning within a median fin that encircles the developing stickleback (stages 28–31)[47]. To examine post-embryonic expression, we designed a knock-in strategy to introduce an enhanced green fluorescent protein (eGFP) reporter gene upstream of the endogenous *HOXD11B* locus using CRISPR–Cas9 (Fig. 1c). The reporter line was generated in an anadromous *Gasterosteus* background from the Little Campbell River (LITC), British Columbia, Canada, a typical three-spine population that migrates between marine and freshwater environments[48]. At stage 19–20, we saw GFP expression in the posterior somites and tailbud, a pattern that recapitulated the *HOXD11B* in situ hybridization results at this embryonic stage (Extended Data Fig. 3c,d). When dorsal spines later form (stage 31), we saw expression in the posterior half of the fish (Fig. 1d), in the dorsal fin fold between the DS2 and DSL, DSL, dorsal fin (DF) (Fig. 1e), the anal fin (AF) and the anal spine (AS). This reporter expression suggests that *HOXD11B* is expressed both in early development and later during dorsal spine formation (a conclusion also supported by RNA sequencing (RNA-seq); see below and Extended Data Fig. 5).

**Fig. 1 | Genetic mapping, expression and role of *HOXD11B* in stickleback dorsal spine development. a**, *Gasterosteus* mapping cross. **b**, QTL scan results for spine number and spine length. *x* axis: *Gasterosteus* chromosomes; *y* axis: LOD score for three- versus four-spine trait (top), length of DS2 (bottom). The QTL peak on chromosome 6 includes the *HOXDB* cluster (gene diagram at the bottom, scale bar, 1 kb). The peak on chromosome 4 includes the *EDA-MSX2A-STC2A* cluster described elsewhere[27,28]. Dashed lines: genome-wide significance thresholds from permutation testing. **c**, Integration of GFP reporter using CRISPR–Cas9 upstream of the endogenous *HOXD11B* locus of low-spine *Gasterosteus*. Plasmid: grey; eGFP: green; basal hsp70 promoter: blue; chromosomal locus: black. Scale bar, 100 bp. TSS, transcription start site. **d**, eGFP expression in posterior half of fish at the stage when the dorsal spines are forming (Swarup stage 31). Scale bar, 1 mm. **e**, Note expression in fin fold between DS2 and DSL, DSL and dorsal fin (DF). Scale bar, 1 mm. **f**, X-ray of uninjected *Gasterosteus* (top) and *Gasterosteus* injected at the single-cell stage with Cas9 and sgRNA targeting the coding region of *HOXD11B* (bottom). Arrows: two blank pterygiophores are often located between DS2 and DSL but only in uninjected fish (insets: two blank pterygiophores in $n = 5$ out of 18 control and $n = 0$ out of 23 injected F0 mutants, two-tailed Fisher's exact test $P = 0.01$). Scale bar, 5 mm. **g**, Length comparisons of dorsal and anal spines. Box and whisker plot: centre line, median; box limits, interquartile range (IQR); whiskers, 1.5× IQR; individual measurements shown as single points (circles: WT; triangles: mutant). *y* axis: residuals after accounting for standard length of fish (Extended Data Fig. 2a). DSL and AS were significantly longer in injected than uninjected fish (two-tailed *t*-test Bonferroni-corrected at $\alpha = 0.05$, $n = 18$ control and $n = 23$ injected, DSL $P_{adj} = 3 \times 10^{-5}$, AS $P_{adj} = 0.02$). DS1 and DS2 lengths were not significantly different.

To determine if *HOXDB* genes are functionally important for stickleback spine patterning, we used CRISPR–Cas9 to target the coding region of *HOXD11B* in typical anadromous low-spine *Gasterosteus* (Little Campbell). Fish in the F0 generation that were mosaic for different mutations in the coding region of *HOXD11B* showed significantly longer DSL compared to their uninjected control siblings (Fig. 1f,g). The anal spine was also significantly longer (Fig. 1f,g). We also saw an effect on the number of pterygiophores, along the dorsal midline (Fig. 1f and Extended Data Fig. 2a). While low-spine *Gasterosteus* develop

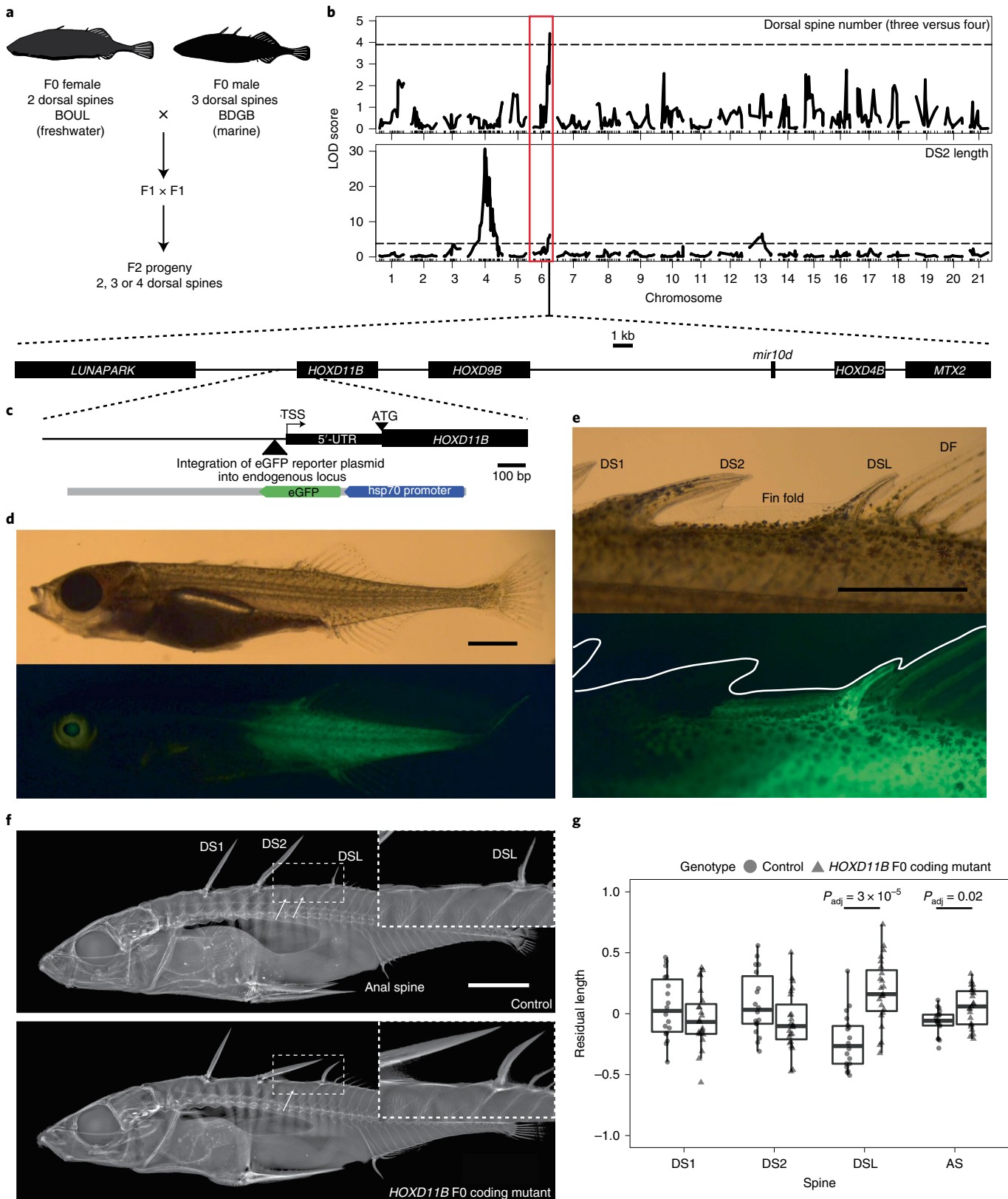

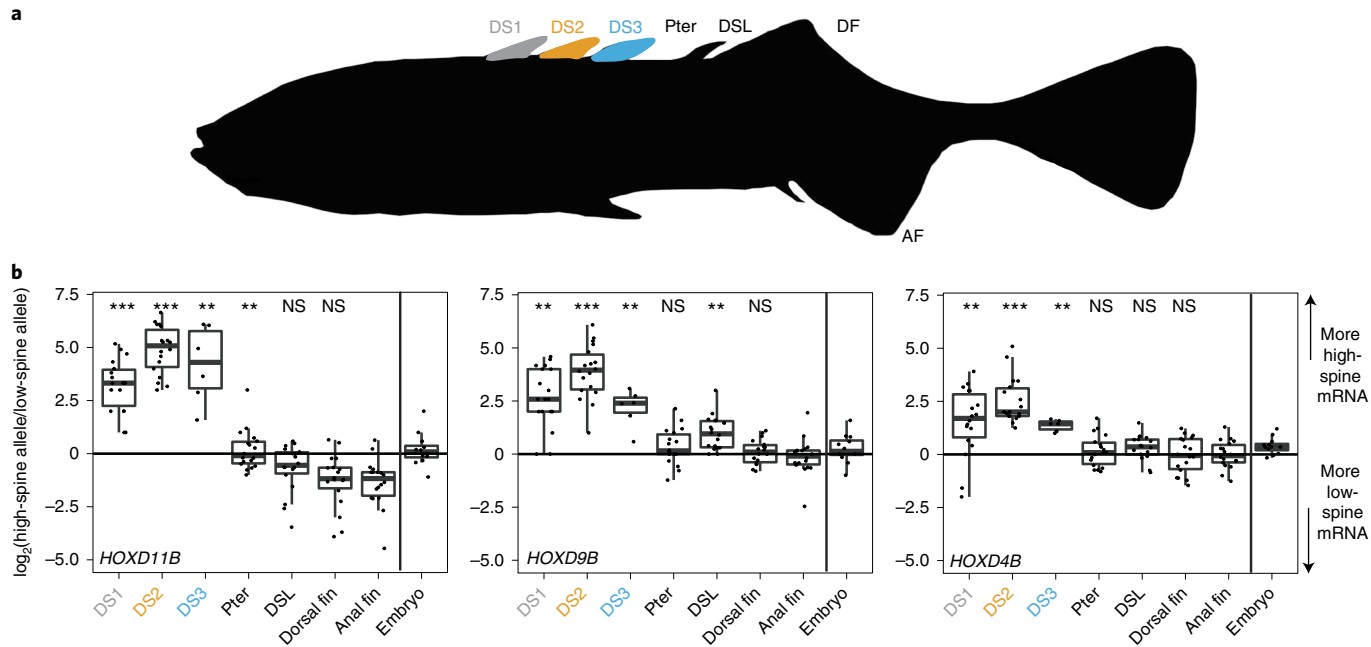

**Fig. 2 | HOXDB genes show cis-acting expression differences in Gasterosteus spines. a**, F1 progeny were generated in a cross between a low-spine and high-spine Gasterosteus and tissues were isolated from up to seven indicated locations (DS1, DS2, DS3 (if present), blank pterygiophore (Pter), DSL, DF and AF) to measure allele-specific gene expression in the fin fold stage. Note DS3 only developed in some F1 progeny, so this location has fewer samples ($n = 6$ for DS3; $n = 18$ for all other tissues). **b**, The box plots show the ratios of high-spine to low-spine allele expression at each of three HOXDB genes. The y axis is the $\log_2$ of the high-spine versus low-spine read ratio at a SNV (black line: equal expression at a $\log_2$ ratio of 0). The x axis shows the seven tissues collected from fin fold stage fish arranged from anterior to posterior, as well as the sample collected from earlier whole embryos (embryo). Centre line, median; box limits, IQR; whiskers, 1.5× IQR; each measurement is represented by a single point. SNVs scored for each dorsal tissue compared to the anal fin: HOXD11B, chrVI:17,756,571; HOXD9B, chrVI:17,764,664; and HOXD4B, chrVI:17,783,616 (gasAcu1-4). Differences were significant in the anterior spines for all three HOXDB genes (DS1: HOXD11B (chrVI:17,756,571) $P = 3 \times 10^{-7}$; HOXD9B (chrVI:17,764,664) $P = 6 \times 10^{-6}$; HOXD4B (chrVI:17,783,616) $P = 1 \times 10^{-5}$; DS2: HOXD11B (chrVI:17,756,571) $P = 3 \times 10^{-7}$; HOXD9B (chrVI:17,764,664) $P = 4 \times 10^{-7}$; HOXD4B (chrVI:17,783,616) $P = 4 \times 10^{-7}$; DS3: HOXD11B (chrVI:17,756,571) $P = 4 \times 10^{-4}$; HOXD9B (chrVI:17,764,664) $P = 8 \times 10^{-4}$; HOXD4B (chrVI:17,783,616) $P = 6 \times 10^{-4}$; all P values by two-tailed Mann–Whitney U-test). **$P \leq 1 \times 10^{-3}$, ***$P \leq 1 \times 10^{-6}$. All alleles with 0 reads have been replaced with 0.5 for graphical representation purposes and statistical analysis. NS, not significant.

one or two blank (non-spine-bearing) pterygiophores between DS2 and DSL, all CRISPR–Cas9 targeted fish developed only one. To further validate these results, we also tested the effect of HOXD11B targeting in a second population (Rabbit Slough (RABS), Alaska). Again, we observed a significant effect on the length of the DSL and AS (Extended Data Fig. 4). There was no effect on spine number in either population. These results show that HOXD11B is functionally important for dorsal skeletal development.

**HOXDB expression is expanded in high-spine Gasterosteus.** To examine whether four-spine/high-spine Gasterosteus fish have cis-acting regulatory changes in HOXDB gene expression, we generated F1 hybrids between low-spine and high-spine stocks and used RNA-seq to look for allele-specific expression patterns detectable even when both alleles were present in the same trans-acting environment. The hybrids were generated by crossing LITC anadromous fish, which predominantly have three dorsal spines ('low-spine'), with a stock descended from the QTL progeny that carry the Boulton HOXDB allele and predominantly show four or five spines ('high-spine'; Methods). In this cross, 77% of 57 F1 hybrids had 3 dorsal spines, 21% had 4, and 1 fish had 5. RNA was isolated from micro-dissections of each dorsal spine (DS1, DS2, DS3 (if present), DSL), blank pterygiophore (Pter), DF and AF at the developing fin fold stage (Fig. 2a), and separately from whole embryos at embryonic stage 19/20.

All three HOXDB genes were expressed in the whole embryo samples from stages 19–20 (Fig. 2b). Reads from RNA-seq were assigned to low- or high-spine HOXDB alleles using exonic single-nucleotide variants (SNVs) that differ between the Little Campbell and Boulton haplotypes (Extended Data Fig. 6). HOXD9B showed no significant allele-specific expression differences at 5 different informative SNVs. HOXD11B showed differences at 3 of 8 SNVs (binomial test $P < 0.01$), and HOXD4B showed differences at 3 of 6 informative SNVs (binomial test $P < 0.001$) (Fig. 3b). Different results for different SNVs may reflect the heterogeneity of expression locations and gene isoforms present in whole embryos. Overall, there were no striking expression differences between the two alleles in embryos.

At the later fin fold stage, we sequenced dissected tissues from 12 three-spined and 6 four-spined F1 individuals. We compared the expression in the dorsal spines and fin to anal fin expression as a control. DS1, DS2 and DS3 showed allele-specific expression differences of all three HOXDB genes. Higher expression was seen from the high-spine parent allele (Fig. 2b). Expression differences were seen in all F1 hybrid siblings, regardless of whether they had a three- or four-spined phenotype. Elevated expression of the high-spine allele was not seen at the Pter, DSL or DF locations (Fig. 2b). In DS1 and DS2, almost all detectable sequence reads for all three HOXDB genes came from the high-spine Gasterosteus allele. This is consistent with the previous patterns observed with the HOXD11B low-spine GFP reporter line, which showed low-spine allele expression at

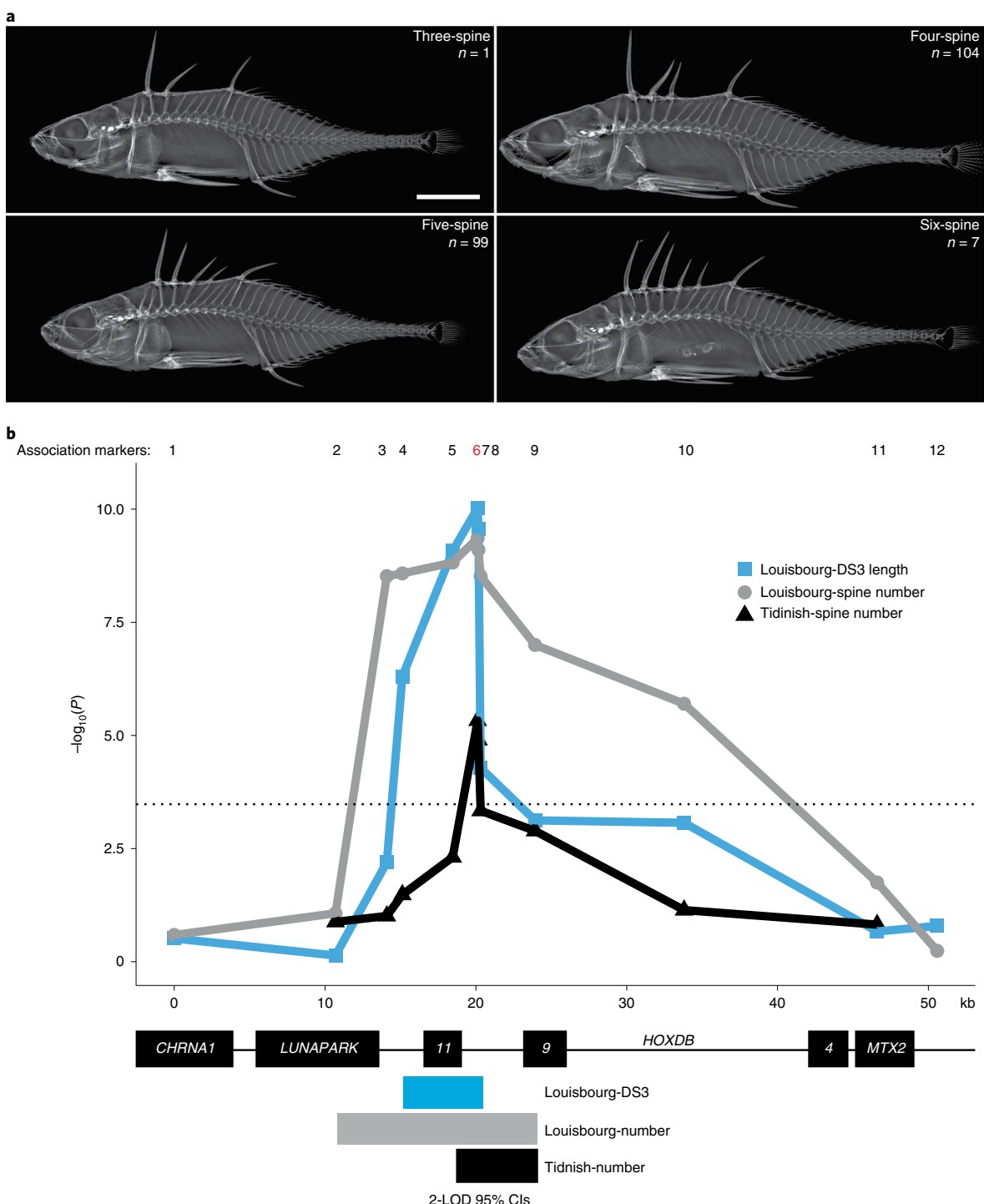

**Fig. 3 | Spine number and DS3 length are associated with the *HOXDB* locus in *Apeltes*. a**, X-rays of *Apeltes quadracus* from Louisbourg Fortress with 3–6 dorsal spines. Scale bar, 5 mm. **b**, Association mapping of *Apeltes* from Louisbourg Fortress (*n* = 211) and Tidnish River 3 (*n* = 121). Both populations show a significant association between spine number and the *HOXDB* locus. *Apeltes* from Louisbourg Fortress were also phenotyped for dorsal spine length and showed a significant association between DS3 length and the *HOXDB* cluster. The markers used are displayed across the top (1–12) and the peak marker (6) is highlighted in red. The dotted line represents the Bonferroni-corrected significance threshold at $\alpha = 0.05$. The 95% confidence intervals (CIs) (2-LOD) for spine number are denoted by the bars at the bottom (grey for Louisbourg, black for Tidnish). The overlapping 95% CI for spine number is approximately 5,750 bp from the third exon of *HOXD11B* to the first exon of *HOXD9B*. The smallest interval shared by both spine number and spine length intervals is approximately 2 kb, including *HOXD11B* exon 3 and part of the intergenic region between *HOXD9B* and *HOXD11B*. Additional anatomical details and association plots for other spine lengths are shown in Extended Data Fig. 7.

posterior, but not anterior, fin fold locations (Fig. 1e). Elevated expression from the high-spine allele led to a significant positive $\log_2$ ratio of high-spine to low-spine expression in each of the dorsal spines when compared to the anal fin (Fig. 2b). Similar results were seen for all SNVs that were scoreable in the 3 *HOXDB* genes (*HOXD11B*, 10 SNVs; *HOXD9B*, 10 SNVs; *HOXD4B*, 8 SNVs).

**HOXDB is associated with spine number and length in *Apeltes*.** To determine if other stickleback genera use the same locus to control dorsal spine patterning, we conducted an association mapping study in *Apeltes quadracus*. As their scientific name suggests, *Apeltes* "quadracus" typically has four dorsal spines. However, multiple wild populations in Canada show more or fewer spines[49] (see Fig. 3 and Extended Data Figs. 7 and 8 for further details of anatomy). *Apeltes* spine number differences are heritable and correlated with ecological conditions[50,51]. We sampled 2 populations in Nova Scotia: 'Louisbourg Fortress' with predominantly 5 dorsal spines (range 3–6; Fig. 3a) and 'Tidnish River 3' with predominantly 4 dorsal spines (range 2–6, Extended Data Fig. 8). We genotyped roughly equal numbers of low-spine (2–4 spines) and high-spine (5–6 spines) individuals across the *HOXDB* locus (Louisbourg $n = 211$ total, 1 three-spine, 104 four-spine, 99 five-spine, 7 six-spine; Tidnish $n = 121$ total, 1 two-spine, 1 three-spine, 59 four-spine, 59 five-spine, 1 six-spine). We observed a highly significant association between spine number and genotypes at two markers located between *HOXD9B* and *HOXD11B* (Fig. 3b, black line). At the peak marker (*AQ-HOXDB_6*), wild fish homozygous for the AA allele had an average of 5.1 spines (s.d. = 0.4) while fish homozygous for the GG allele had an average of 4.2 spines (s.d. = 0.5).

We also measured spine length to test whether, as in *Gasterosteus*, the *Apeltes HOXDB* cluster was associated with spine length changes in the Louisbourg fish. We numbered spines similarly to *Gasterosteus*, with a three-spine *Apeltes* having from anterior to posterior DS1, DS2 and DSL, and a six-spine *Apeltes* having DS1, DS2, DS3, DS4, DS5 and DSL (Extended Data Fig. 7a). Only DS3 length was strongly associated with genotypes in the *HOXDB* region (Fig. 3b, blue line and Extended Data Fig. 7). The genotype at the peak marker (*AQ-HOXDB_6*) explained 22% of the overall variance in DS3 length of wild-caught fish.

The minimal genomic interval shared by both spine number and length associations was approximately 2 kilobases (kb), including *HOXD11B* exon 3 and part of the intergenic region between *HOXD9B* and *HOXD11B* (Fig. 3b). Based on whole-genome DNA sequencing from Louisbourg ($n = 2$) and RNA-seq data ($n = 14$) (Methods), no sequence variation was found in the protein-coding regions of *HOXD11B* or *HOXD9B*. The peak marker for both associations was a change of two adjacent intergenic bases from GG to AA. Together, these results suggest that increased spine number and increased DS3 length in some *Apeltes* probably arise from a regulatory difference in the non-coding interval between *HOXD9B* and *HOXD11B*.

**Apeltes HOXDB genes show *cis*-regulatory spine differences.** To test for *cis*-acting regulatory differences in *Apeltes HOXDB* genes, we generated F1 hybrids with contrasting *Apeltes* haplotypes in the key genomic interval and carried out RNA-seq on spines, blank pterygiophores and dorsal and anal fins at the fin fold stage (Fig. 4a). While there were no sequence differences in the

protein-coding portions of the *HOXDB* genes, the 3′ untranslated regions (UTRs) of *HOXD9B* and *HOXD11B* had variants that could be used to determine the expression level coming from the genotypes associated with low-spine (L) or high-spine number (H) in the association study (Fig. 4b). In F1 fish carrying one L haplotype and one H haplotype, the L haplotype *HOXD9B* and *HOXD11B* genes had significantly higher expression (Fig. 4c). Differences were most pronounced in DS3, the same spine whose overall length was associated with *HOXDB* genotypes (Fig. 4c).

Some F1 fish generated for the allele-specific expression experiment carried both an H haplotype and a recombinant haplotype that we termed the low-high-recombinant (LHR) haplotype (Fig. 4b). These fish showed an allele-specific expression pattern similar to fish heterozygous for an H and L haplotype, with more expression of *HOXD9B* and *HOXD11B* coming from the LHR haplotype (Fig. 4d). In contrast, fish heterozygous for the LHR and L haplotypes showed no significant difference in *HOXD9B* expression (Fig. 4e). Thus, at a gene expression level, the LHR haplotype behaved more like the L haplotype than the H haplotype. Similarly, at the phenotypic level, F1 individuals heterozygous for the L and LHR haplotypes typically had low spine numbers, resembling fish homozygous for L haplotypes, while fish homozygous for H haplotypes had higher spine numbers (L/L fish: 16 out of 16 with 3 or 4 spines; L/LHR fish: 15 out of 18 with 3 or 4 spines, 3 out of 18 with 5 spines; H/H fish: 16 out of 16 with 5 or 6 spines; two-tailed Fisher's exact test, $P = 9 \times 10^{-11}$; post-hoc pairwise Fisher's exact test, Bonferroni-corrected at $\alpha = 0.05$, L/L versus L/LHR $P_{adj} = 0.7$; L/LHR versus H/H $P_{adj} = 1 \times 10^{-6}$; L/L versus H/H $P_{adj} = 1 \times 10^{-8}$). These results suggest that the key genomic interval controlling both gene expression differences and phenotypic differences between the L/LHR and H haplotypes maps to the minimal approximate 5 kb region shared between the L and LHR haplotypes (pink region on the left of Fig. 4b).

**Genomic changes in a *Gasterosteus* and *Apeltes* spine enhancer.** To search for *cis*-regulatory sequences contributing to *HOXDB* expression variation, we looked for conserved non-coding sequences and open chromatin domains located in the minimal interval defined by the association and gene expression studies. This identified 1 approximately 500 base pair (bp) region (Fig. 5a) found in both *Apeltes* and *Gasterosteus* that is conserved by phastCons alignment to Tetraodon, Medaka, and Fugu[52]. This small conserved region contained the peak scoring marker in the *Apeltes* association study (two adjacent bases changed from GG to AA) (Fig. 5c). This conserved non-coding region also corresponds to a region of open chromatin in Medaka embryos at stages equivalent to those where we see embryonic *HOXDB* expression in sticklebacks[53].

We cloned the *Apeltes* region from both the L and H haplotypes (611 bp in L; 587 bp in H) and tested whether the sequences could drive GFP reporter expression in transgenic enhancer assays. Because *Apeltes* have very small clutch sizes, constructs were

---

**Fig. 4 | *HOXDB* genes show *cis*-acting expression differences in *Apeltes* spines. a**, Outline of *Apeltes* fin fold stage fry. Tissues were isolated from DS1, DS2, DS3, DS4, Pter, DSL, DF and AF to measure allele-specific gene expression in F1 hybrids. (DS4 only developed in some progeny.) **b**, Three *HOXDB* haplotypes segregating in cross. Black lines: association mapping markers. Pink: regions with genotypes associated with low-spine phenotypes in Fig. 3b. Yellow: regions with genotypes associated with high-spine phenotypes in Fig. 3b. Lighter shading: regions where marker association is unknown but DNA variants are shared between haplotypes of the same colour. **c**, Box plots showing allele-specific expression ratios in all tissues dissected from fry heterozygous for the H and L haplotypes ($n = 4$). All box and whisker plots: centre line, median; box limits, IQR; whiskers, 1.5× IQR; each measurement is represented by a single point. Reads from DS3, DS4, Pter, DSL and DF were compared to reads from AF to determine significance (DS3: *HOXD9B* (chr06:16,028,519) $P = 3 \times 10^{-7}$; *HOXD11B* (chr06:16,020,516) $P = 9 \times 10^{-4}$, two-tailed Fisher's exact test). DS1 and DS2 were not assessed because read counts were too low (Extended Data Fig. 5). **d**, Box plots showing allele-specific expression ratios in all the tissues dissected from fry heterozygous for LHR and H haplotypes ($n = 3$). Allele-specific expression was seen in DS3 compared to anal fin for both *HOXD11B* and *HOXD9B* (DS3: *HOXD9B* (chr06:16,027,923) $P = 9 \times 10^{-8}$; *HOXD11B* (chr06:16,020,516) $P = 1 \times 10^{-3}$, two-tailed Fisher's exact test). **e**, Box plot showing allele-specific expression ratios in all tissues dissected from fry heterozygous for L and LHR haplotypes ($n = 4$). Only *HOXD9B* is shown because *HOXD11B* lacks informative variants between L and LHR haplotypes (DS3: *HOXD9B* (chr06:16,027,923) $P = 0.08$, two-tailed Fisher's exact test). **$P \le 1 \times 10^{-3}$, ***$P \le 1 \times 10^{-6}$.

injected into *Gasterosteus* to obtain sufficient transgenic embryos for analysis. The approximately 600 bp non-coding constructs both drove expression at embryonic time points in the tail of transgenic

embryos in a similar pattern to that seen in the in situ hybridizations for *HOXD9B* and *HOXD11B* (Fig. 5b left and Extended Data Fig. 3b,c). At later time points, the conserved non-coding regions

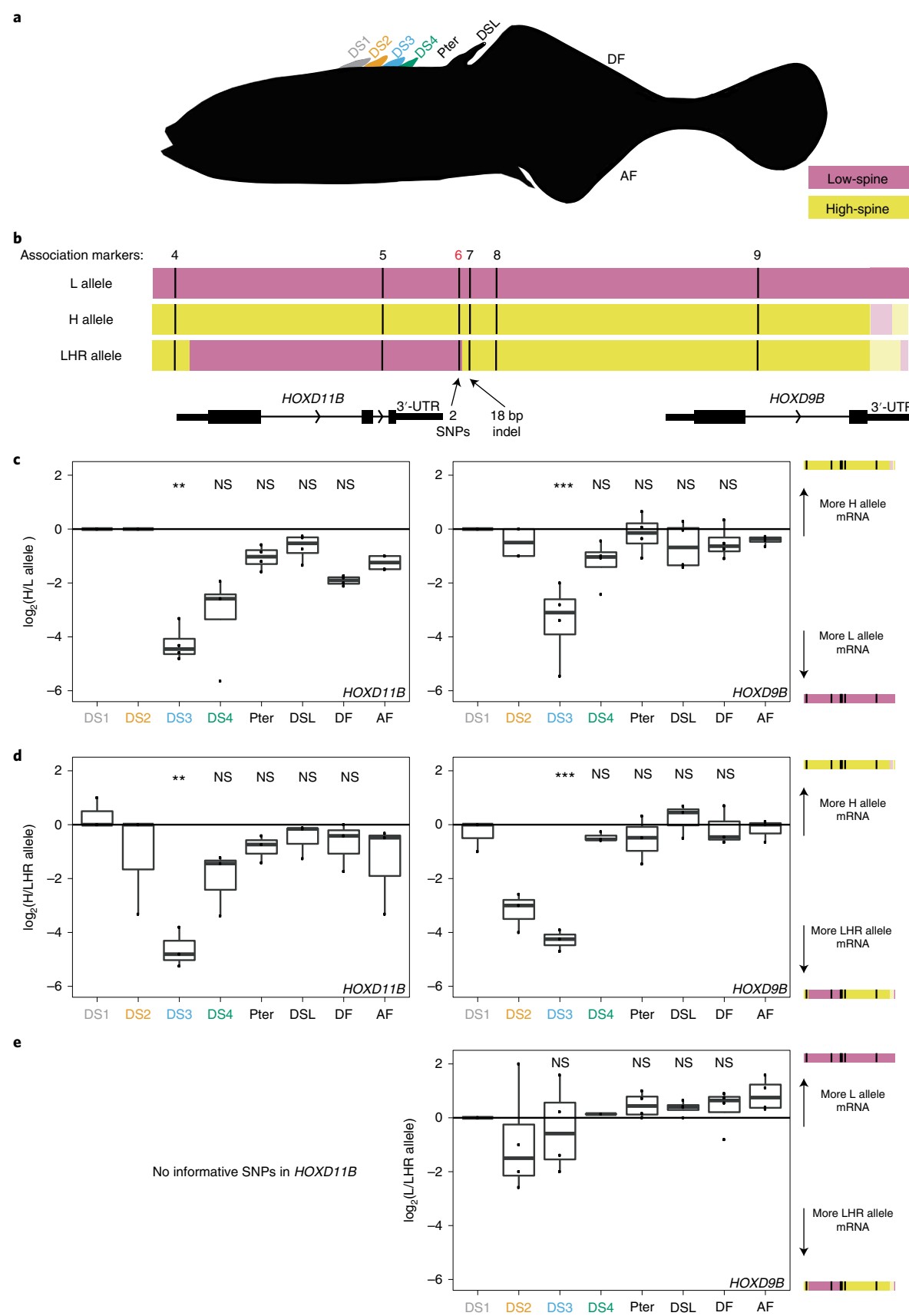

drove expression in the dorsal, caudal and anal fins; dorsal and pelvic spines; and the posterior of the fish (Fig. 5b, right). Similar patterns were driven by both the L- and H-type *Apeltes* constructs (Fig. 5c), although we note that differences in both strength and patterns of expression can be difficult to detect in mosaic transgenic fish resulting from random *Tol2* integration (L 611 bp region: $n = 22$ transgenics with bilateral green eyes; $n = 6$ out of 22 pectoral fin, $n = 8$ out of 22 pelvis, $n = 9$ out of 22 dorsal spines, $n = 11$ out of 22 dorsal fin, $n = 9$ out of 22 anal fin, $n = 10$ out of 22 posterior muscle; H 587 bp region: $n = 19$ with bilateral green eyes; $n = 2$ out of 19 pectoral fin, $n = 4$ out of 19 pelvis, $n = 9$ out of 19 dorsal spines, $n = 11$ out of 19 dorsal fin, $n = 11$ out of 19 anal fin, $n = 10$ out of 19 posterior muscle (Fisher's exact test at all sites: $P_{adj} = 1$)). Given the consistent expression patterns seen in both tail buds and later axial structures of transgenic fish, we refer to the approximately 600 bp conserved intergenic sequence as an axial enhancer (*AxE*) of the *HOXDB* locus.

Although *AxE* sequences are conserved between *Apeltes* and typical *Gasterosteus*, we were unable to amplify the *AxE* region from the Boulton high-spine allele in the *Gasterosteus* QTL cross. We used PacBio long-read sequencing to identify the intergenic region between *HOXD9B* and *HOXD11B* from the Boulton high-spine allele. The sequenced region shows major structural changes, including a deletion that removes almost all of the *AxE* and the presence of two transposable elements not present at this location in the low-spine reference genome[43]: a long interspersed nuclear element (LINE) (L2–5_GA) element and an endogenous retrovirus (ERV1-6_GA-I) (Fig. 5a). The LINE element is approximately 1 kb and also present in additional *Gasterosteus* populations in the Pacific Northwest (sequencing data from[54]). When the LINE element was detected in other populations, it was not associated with the *AxE* sequence deletion seen in Boulton. The ERV insertion was approximately 11 kb containing open reading frames for an envelope and Gag-Pol proteins, flanked by approximately 1 kb long terminal repeats (LTRs). Junction sequences for this retroviral insertion near *AxE* were not found in the sequenced genomes of over 200 *Gasterosteus* from different populations[43,54]. Thus, the Boulton high-spine allele shows both the nearly complete loss of *AxE* and the addition of new sequences.

To determine if loss of the *AxE* sequence alone was sufficient to recapitulate the phenotypic effect of a higher spine number and shorter DS2 length in *Gasterosteus*, we deleted the region in low-spine *Gasterosteus* using CRISPR targeting. In mosaic F0 founder fish, no significant effects on spine number were detected. However, DSL and AS were significantly longer in the F0 injected mutants compared to their control siblings (Extended Data Fig. 9). Both the spines affected and the direction of phenotypic effects resembled the phenotypes seen when targeting the *HOXD11B* protein-coding region. These results suggest that the *AxE* region is required for normal spine length patterning in *Gasterosteus* but that additional sequence changes probably contribute to the spine number phenotypes linked to the region.

To test whether any of the additional transposable element sequences in the Boulton high-spine allele might contribute new enhancer activities, we tested whether the approximately 1 kb LINE or the approximately 1 kb LTR of the ERV could drive GFP reporter gene expression in transgenic enhancer assays compared to an empty vector control. While the empty vector and the LINE did not drive expression at an early fin fold stage (Swarup stage 29), the LTR drove expression in the dorsal fin fold, anal fin, posterior muscle, heart and gills (empty vector: $n = 17$ transgenics with bilateral green eyes; $n = 2$ out of 17 whole-body, $n = 1$ out of 17 heart, $n = 0$ out of 17 posterior muscle, $n = 0$ out of 17 dorsal fin fold, $n = 0$ out of 17 anal fin fold, $n = 0$ out of 17 gill, $n = 0$ out of 17 caudal fin; LINE: $n = 13$ transgenics with bilateral green eyes; $n = 0$ out of 13 whole-body, $n = 0$ out of 13 heart, $n = 0$ out of 13 posterior muscle, $n = 0$ out of 13 dorsal fin fold, $n = 0$ out of 13 anal fin fold, $n = 0$ out of 13 gill, $n = 0$ out of 13 caudal fin; LTR $n = 28$ transgenics with bilateral green eyes; $n = 3$ out of 28 whole-body, $n = 10$ out of 28 heart, $n = 16$ out of 28 posterior muscle, $n = 13$ out of 28 dorsal fin fold, $n = 12$ out of 28 anal fin fold, $n = 4$ out of 28 gill, $n = 16$ out of 28 caudal fin; Fisher's exact test empty vector versus LTR, whole-body $P_{adj} = 1$, heart $P_{adj} = 0.2$, posterior muscle $P_{adj} = 4 \times 10^{-4}$, dorsal fin fold $P_{adj} = 4 \times 10^{-3}$, anal fin fold $P_{adj} = 7 \times 10^{-3}$, gill $P_{adj} = 1$ and caudal fin $P_{adj} = 4 \times 10^{-4}$). Thus, the ERV insertion that is unique to the Boulton allele includes new *cis*-regulatory enhancer sequences. Additional regulatory sequences may be present in the rest of the insertion or surrounding region.

## Discussion

Vigorous historical debates have existed about whether mutations in homeotic genes are the likely basis of common morphological changes seen in wild animals. Most laboratory or human-selected mutations in genes are deleterious. In addition, transposable element insertions are strikingly depleted at *Hox* loci, an effect attributed to the likely deleterious consequences of making substantial regulatory changes in genes essential for development and survival[55,56]. On the other hand, the diversity of *Hox* cluster number, composition and expression patterns, and the powerful effects of *Hox* genes on many phenotypes in laboratory models, have made the genes often-cited candidates for the molecular basis of phenotypic differences between wild species, including sticklebacks[45,46]. *Cis*-acting regulatory differences at *Hox* loci clearly underlie evolutionary differences in trichome and pigmentation patterns in insects, but the underlying molecular changes are not known[20,21]. Our studies show that independent regulatory changes have occurred in the *HOXDB* locus of *Gasterosteus* and *Apeltes*, providing a compelling example of *cis*-acting variation in *Hox* genes linked to the evolution of new axial skeletal patterns in wild vertebrate species.

**Adaptive significance of dorsal spine number and length.** Dorsal spines in sticklebacks play an important role in predator defence. Long spines increase the effective cross-sectional diameter of sticklebacks[57] and can provide a survival advantage against gape-limited

**Fig. 5 | The genomic region between *HOXD11B* and *HOXD9B* contains a conserved *AxE* showing sequence changes in both *Gasterosteus* and *Apeltes*.**
**a**, The protein-coding exons of *HOXD11B* and *HOXD9B* are shown in *Gasterosteus* (*gasAcu*1) genomic coordinates. Sequence conservation: phastCons conserved sequence regions identified in exons and in an approximately 500 bp intergenic region from comparisons between fish genomes. The conserved non-coding region overlaps an assay for transposase-accessible chromatin using sequencing (ATAC-seq) peak from Medaka embryonic stage 19 (ref. [53]) and partially overlaps the genomic intervals defined by spine phenotype and RNA expression changes in *Apeltes*. In high-spine *Gasterosteus*, the conserved region *AxE* is deleted (as indicated by a black line between the two grey boxes) and ERV and LINE sequences are inserted (in red and yellow, respectively). **b**, Approximately 600 bp *AxE* regions from low-spine and high-spine *Apeltes* were cloned into a *Tol2* GFP expression construct and injected into *Gasterosteus* embryos. Both versions drove expression in the tailbud of embryos (left) and the fin fold, spines, and dorsal, anal and caudal fins of stage 31 fry (right), confirming that the region acts as an enhancer. Scale bar, 1 mm. **c**, There are four sequence differences in the *AxE* region of high- and low-spine *Apeltes* alleles: one microsatellite variation; an 18 bp indel; a SNP; and 2 adjacent SNPs. Only the single and two adjacent SNPs are within the region implicated by *Apeltes* recombination and RNA-seq differences (pink bar).

predators[58]. Prominent spines also provide holdfasts for grappling insect predators and may therefore increase the risk of predation by macroinvertebrates[39,59]. Different predation regimes may thus favour either increased or decreased spine lengths and numbers. The intensity of bird, fish and insect predation varies across

locations, years and seasons, contributing to a range of spine phenotypes in natural stickleback populations[60–63].

Boulton is an extensively studied population where fish typically show two or three dorsal spines[39]. Detailed seasonal and longitudinal surveys showed that lower spine numbers in Boulton fish are

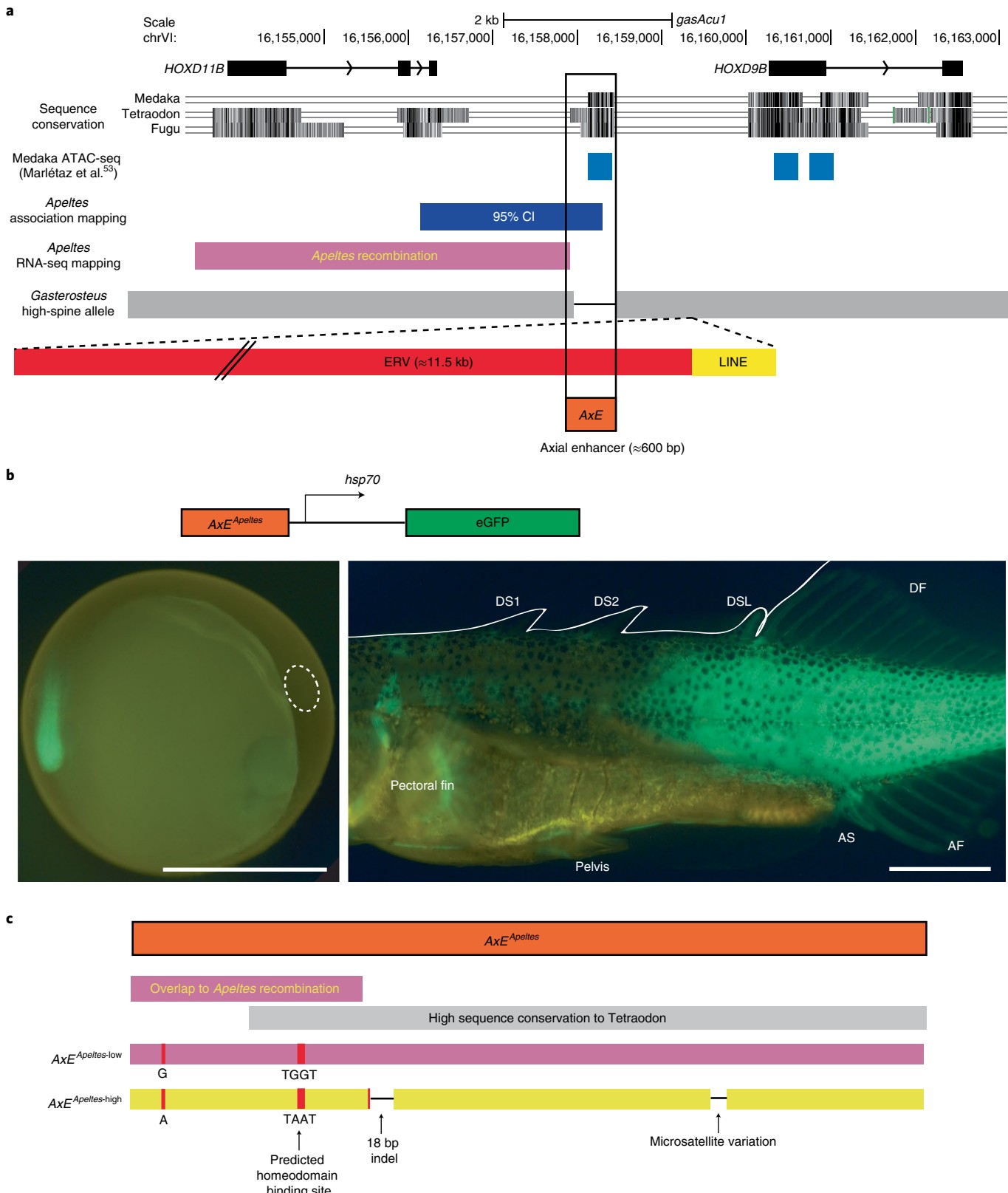

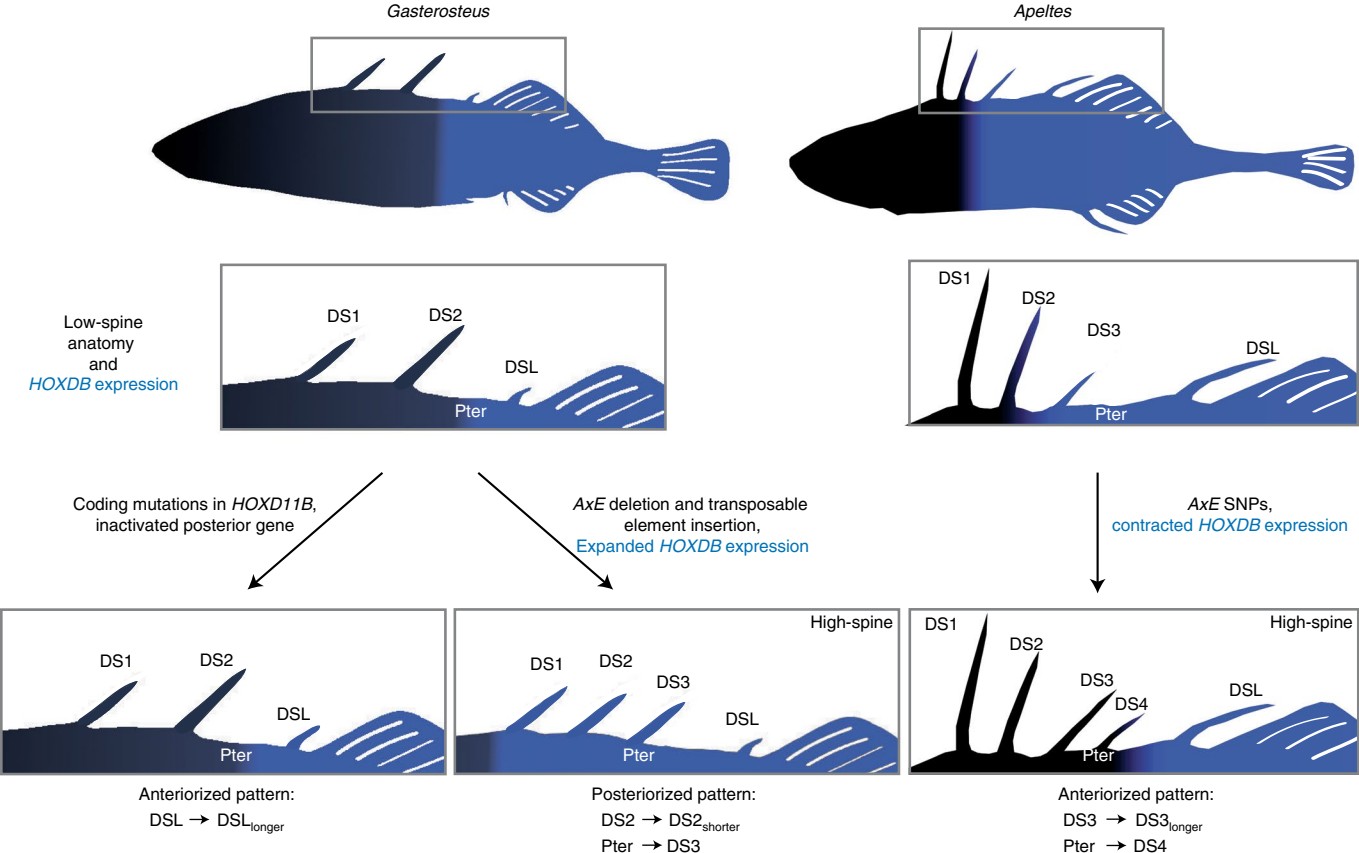

**Fig. 6 | Model of dorsal spine pattern changes resulting from coding and expression differences in laboratory and wild stickleback populations.** Typical *Gasterosteus aculeatus* (three-spine sticklebacks) have a convex pattern of spine lengths where the middle/DS2 spine is the longest). All three *HOXDB* genes have no or low expression in the anterior, DS1 and the strongest expression in the posterior DSL regions. CRISPR-induced coding region mutations in *HOXD11B* lengthen DSL. This is expected since the loss of a posterior *Hox* gene typically results in the anteriorization of structures; in this case, DSL is longer and thus looks more like DS2 (left). In contrast, the naturally occurring Boulton high-spine allele has expanded *HOXDB* gene expression anteriorly, such that all the *HOXDB* genes are expressed in all of the spines. Corresponding morphological effects seen in the QTL cross (shortening of DS2 and addition of a new spine on a blank pterygiophore between DS2 and DSL) are both consistent with posteriorization of dorsal midline structures (middle bottom). These morphological and gene expression changes are linked to *cis*-regulatory changes at the *HOXDB* locus, including the deletion of an enhancer (*AxE*) and insertion of two transposable elements. Typical *Apeltes quadracus* fish have a concave pattern of spine lengths (middle/DS3 spine the shortest). In low-spine *Apeltes*, *HOXDB* genes are strongly expressed from DS3 to the posterior. In high-spine *Apeltes*, the anterior boundary of *HOXDB* expression is shifted to the posterior and thus DS3 has lower *HOXDB* expression. Loss of expression of *HOXDB* genes is associated with lengthening of DS3 and the formation of a new spine on a pterygiophore between DS3 and DSL, both consistent with anteriorization of dorsal midline identities.

correlated with a higher intensity of insect predation and higher spine numbers with a higher intensity of bird predation[60,64]. Because fish with four dorsal spines have not been seen in over 20,000 wild-caught Boulton fish, the occurrence of four-spine sticklebacks in the Boulton × Bodega Bay F2 laboratory cross is a transgressive phenotype[65] that emerges when Boulton alleles are inherited on a mixed genetic background. We note, however, that the Boulton *HOXDB* region is also linked to shortening of DS2 in the QTL cross. We hypothesize that a *HOXDB* allele probably evolved in this population for its contributions to reduced DS2 length via a posteriorizing mechanism (Fig. 6). In the Boulton genetic background, this posteriorizing effect may be sufficient to change DS length without inducing formation of a new spine on a blank pterygiophore, while in the mixed genetic background of the QTL cross, the posteriorizing tendency of the Boulton allele leads to both spine length and number changes.

Although *Apeltes* sticklebacks typically develop four dorsal spines, many "*quadracus*" populations in Canada have predominantly three or five spines[49,50]. In an extensive comparison of *Apeltes* spine numbers and environmental variables across 570 locations,

Blouw and Hagen[66–68] found that increased spine number was correlated with the presence of predatory fish, while decreased spine number was correlated with more and denser vegetation. To study the possible adaptive value of spine number differences, Blouw and Hagan exposed mixed populations of four- and five-spine *Apeltes* to predatory fish and measured differential survival of spine morphs when half of the sticklebacks had been eaten[67]. When vegetation was present, predation was non-selective; however, when vegetation was absent, five-spine fish were less likely to be eaten by perch and trout. Consistent with the experimental predation experiments and known ecological correlations, the stomachs of wild-caught trout contained more four- than five-spine fish, while the stomachs of herons contained more five- than four-spine fish[67]. Thus, the dorsal spines in sticklebacks provide an excellent example of a prominent adaptive structure in vertebrates that evolves in response to different predation regimes in natural environments and diversifies in part through repeated regulatory changes in *Hox* genes.

Although spine phenotypes are clearly adaptive in sticklebacks, we note that we have not yet established whether the specific *HOXDB* alleles identified in the Boulton and Nova Scotia

populations have been subjected to positive selection. Multiple dorsal spine phenotypes have been present in the Boulton, Louisbourg and Tidnish populations for decades, probably due to long-term balancing selection in the face of fluctuating predation regimes[49,60,67]. Long-term balancing selection is more difficult to detect at the molecular level than selective sweeps of a favoured allele[69]. However, recent experiments have successfully monitored short-term changes in the frequency of alleles of interest in sticklebacks after changes in environmental conditions[54,70] or experimental exposure to predators[59]. In the future, it will be interesting to extend such studies to differential survival of particular *Hox* alleles and dorsal spine phenotypes, using either the alternative alleles we have identified in this study from natural *Gasterosteus* and *Apeltes* populations or CRISPR-edited sticklebacks engineered to carry particular *HOXDB* sequence variants.

**Hox genes and dorsal midline skeletal patterns.** *Hox* genes are well known for controlling the identity of structures in repeating series, including body segments in insects, somite fates in vertebrates, digit identities in limbs and rhombomere segments in the hindbrain[10]. We propose that changes in dorsal spines and pterygiophores of fish also represent identity transformations within the dorsal midline. Many *Hox* changes are governed by 'posterior prevalence', where the posterior-most, highest-numbered *Hox* gene expressed in a given region generally controls the fate of that region[71,72]. Therefore, when posterior gene expression expands, the regions with expanded expression generally acquire a more posterior fate. Conversely, when activity of a *Hox* gene is lost from a region, that region typically acquires a more anterior fate.

Our RNA-seq studies show that many *Hox* genes in both *Gasterosteus* and *Apeltes* are expressed in the dorsal spines or fins of developing sticklebacks, with the exception of *Hox* PG1, some *Hox* PG13 genes, and all *HOXBB* cluster genes (Extended Data Fig. 5). Several *Hox* genes show strong differential expression across different spines and pterygiophores (including *HOXDB* genes), suggesting that morphological fates in the dorsal midline are probably influenced by the combined expression of multiple genes.

The *Gasterosteus* high-spine allele that causes expanded expression of the *HOXDB* genes would be predicted to result in a posteriorization of the regions gaining expression. As expected, this allele is associated with both a blank pterygiophore developing a spine (consistent with a shift to a more posterior identity like DSL) and the shortening of DS2 (consistent with what is typically the longest spine becoming more like the shortest, DSL). Conversely, knocking down *HOXD11B* expression by CRISPR–Cas9 is predicted to result in anteriorization of structures; the increased length of the DSL that we observe is consistent with a shift of DSL to a more anterior and therefore longer spine fate (Fig. 6).

The *Apeltes* H allele that shows reduced *HOXD9B* and *HOXD11B* gene expression in DS3 is associated with both increased spine number and a longer DS3. The number and length phenotypes can both be interpreted as transformations to a more anterior fate. In this model, the appearance of a fifth spine on a normally blank pterygiophore could be explained by partial transformation to a more anterior, spine-bearing fate. Anterior spines are normally longer than posterior spines in *Apeltes*, so the increased length of DS3 is also consistent with an anterior transformation (Fig. 6).

**Independent changes in HOXDB cis-regulatory elements.** Our allele-specific expression experiments show that the changes in *HOXDB* expression we see in sticklebacks are due to *cis*-acting regulatory differences linked to the *Hox* genes themselves, rather than secondary consequences of changes in unknown *trans*-regulatory factors. Our mapping, association, and transgenic experiments have identified a particular *cis*-acting enhancer region located between *HOXD9B* and *HOXD11B* that can recapitulate axial expression

patterns and shows independent sequence changes in *Gasterosteus* and *Apeltes* with different spine numbers. In *Apeltes*, the most likely sequence difference mediating changes of *HOXD9B* and *HOXD11B* expression are two adjacent SNPs (marker: *AQ*-6) in *AxE*. These SNPs convert a TGGT sequence in the L allele to TAAT in the H allele. They represent the peak marker scored by association mapping and also map within the minimal recombination interval that controls H versus L and LHR *cis*-acting expression differences seen in F1 hybrids (Fig. 5c). Both the TAAT change and a nearby 18 bp indel, which represents the second highest scoring marker (*AQ*-7) in the association mapping, are found in the high-spine fish of two populations on opposite coasts of Nova Scotia. Thus, repeated evolution of different high-spine *Apeltes* populations probably takes place through a shared underlying molecular haplotype at the *Hox* locus rather than independent mutations in these different populations. A similar allele sharing process underlies recurrent evolution of a variety of other phenotypic traits in both sticklebacks and other organisms[35,73,74]. We note that the derived TAAT sequence in the *Apeltes* high-spine allele creates a predicted core binding motif for a homeodomain protein. Previous studies in *Drosophila* and other organisms have shown that *Hox* genes can autoregulate in positive and negative feedback loops[75–77]. We hypothesize that the creation of a new putative homeodomain binding site located between the *HOXD9B* and *HOXD11B* genes may contribute to the decreased *HOXDB* expression observed with the H allele. Note that we have not been able to recapitulate the altered expression patterns using *AxE* transgenic reporter constructs integrated at random locations in the genome. Because endogenous *Hox* expression patterns are likely controlled by interactions between multiple long-distance control elements and surrounding topological domains[78,79], the most accurate functional tests of the phenotypic effects of mutations will come from scoring those changes at their correct genomic position. Future advances in genome editing may eventually make it possible to recreate or revert the TGGT and TAAT sequence change at the endogenous *HOXDB* locus in sticklebacks and further test whether these two adjacent base pair changes are sufficient to alter *Hox* gene expression and spine length or number.

In *Gasterosteus*, the *AxE* enhancer has been deleted from the Boulton *HOXDB* high-spine allele and replaced with two transposable elements, one ERV and one LINE. Removing the endogenous *AxE* enhancer by CRISPR–Cas9 does not lead to spine number and DS2 phenotypes but it does recapitulate the DSL length changes seen by targeting the *HOXD11B* coding region. The LTRs found in ERVs can act as enhancers[80], including the LTR sequences inserted in the Boulton allele. We hypothesize that the large insertion in the Boulton allele underlies broader expression in the dorsal spines and the other phenotypic consequences of the Boulton high-spine allele.

**Repeated regulatory changes in morphological evolution.** A long-standing question in evolutionary biology is whether the same genetic mechanisms are used repeatedly to evolve similar traits in different populations and species. Although *Gasterosteus* and *Apeltes* last shared a common ancestor over 16 million years ago[30], our data show that both stickleback groups have independent *cis*-regulatory changes in *HOXDB*, which are linked to new dorsal spine patterns in recently evolved populations. The types of mutations made in the *AxE* regulatory region are clearly distinct and the naturally occurring *Gasterosteus* and *Apeltes* H alleles lead to contrasting *HOXDB* expression changes. Interestingly, the *HOXD* locus is also used repeatedly during horn evolution in mammals. The *HOXD* region shows accelerated evolution and insertion of a new retroviral element in the diverse clade of species with horns and antlers[81]. In addition, rare four-horned sheep and goats have recently been shown to have independent mutations in the *HOXD* locus, ranging from a 4 bp mutation that alters splicing to a large (approximately 500 kb) deletion that is lethal when homozygous[82–84].

The fish and mammalian results support a growing body of literature that has found repeated use of the same loci underlying similar traits, even though the direction of effect of gene expression and mutational mechanism are often different[74].

While both of our examples of spine variation in recently diverged populations of *Gasterosteus* and *Apeltes* involve *cis*-regulatory changes, *Hox* coding region mutations may also contribute to diversification of spine patterns over a wider phylogenetic scale. For example, the Gasterosteidae family can be separated into five different genera of predominantly three-spine, four-spine, five-spine, nine-spine and fifteen-spine sticklebacks (*Gasterosteus*, *Apeltes*, *Culaea*, *Pungitius* and *Spinachia*, respectively). We note that the coding region of *HOXD11B* shows a high rate of non-synonymous to synonymous substitutions across the stickleback family; the dN/dS ratio is greater than 1.0 for comparisons between *Apeltes* and *Gasterosteus* (Extended Data Fig. 10). This suggests that changes in *HOXD11B* coding regions have likely been under positive selection during the divergence of *Apeltes* and *Gasterosteus*, perhaps contributing to the distinctive patterns of spine length and number that are characteristic of these two genera.

Spiny-rayed fish are among the most successful vertebrates, making up nearly a third of extant vertebrate species. The lengths and numbers of spines show remarkable diversity across Acanthomorpha, including elaborate modifications that have evolved for defence, camouflage, luring prey or swimming biomechanics[24]. Our results show that changes in the dorsal spine patterns of wild fish species have evolved in part through genetic changes in *Hox* genes. Based on the recurrent use of the same locus for spine evolution in different stickleback species, we hypothesize that repeated mutations in *Hox* genes may also underlie other interesting changes that have evolved in the axial skeletal patterns of many other wild fish and animal species.

## Methods

**Ethical compliance.** We complied with all relevant ethical regulations during this study.

**Stickleback care.** Sticklebacks were captured using minnow traps or dip and seine nets. Populations and their Global Positioning System (GPS) coordinates are in Supplementary Table 1. All sticklebacks were treated in accordance with the Guide for the Care and Use of Laboratory Animals of the National Institutes of Health (NIH), using protocols approved by the Institutional Animal Care and Use Committee of Stanford University (protocol no. 13834), in animal facilities accredited by the Association for Assessment and Accreditation of Laboratory Animal Care International.

**DNA extraction.** DNA was isolated from fins by incubating in lysis buffer (10 mM Tris, pH 8, 100 mM NaCl, 10 mM EDTA, 0.5% SDS, Proteinase K (333 µg ml$^{-1}$); catalogue no. P8107S; New England Biolabs)) at 55 °C for 4 h to overnight, extracting with phenol:choloroform:isoamyl alcohol 25:24:1 (catalogue no. P3803; Sigma-Aldrich) in phase lock tubes (MaXtract High Density, catalogue no. 129056; QIAGEN), ethanol precipitating overnight and resuspending in TE Buffer (10 mM Tris, 1 mM EDTA, pH 8).

**QTL mapping.** A wild-caught female from Boulton was crossed by in vitro fertilization to a marine male stickleback from Bodega Bay. F1 progeny were raised to adulthood in the laboratory in 29 gal aquariums in reverse osmosis-purified water with 3.5 ppt Instant Ocean salt and intercrossed to generate multiple F2 families. Sperm from F1 males were cryopreserved[85], so single males could be crossed multiple times. F2 progeny fish were raised in the laboratory for 1 year, euthanized with 200 mg l$^{-1}$ tricaine methanesulfonate (Abbreviated New Animal Drug Application no. 200-226; Western Chemical) buffered to pH 7 with sodium bicarbonate and preserved in 70% ethanol.

Fish DNA samples were genotyped using an Illumina GoldenGate genotyping array with 1,536 features[40]. Intensity data were processed using GenomeStudio v.2011. Genotype clusters were inspected and adjusted manually and uninformative or low-intensity SNPs were excluded from downstream analysis. Phasing and linkage map construction were performed with TMAP version 0.6[86]. The linkage map and phased genotype data were then loaded into R/qtl version 1.26.0[87] and filtered to remove fish with fewer than 600 genotype calls and markers with fewer than 300 calls. A final map was generated with 343 F2s and 452 markers.

*Gasterosteus* anatomical traits and landmarks are diagrammed in Extended Data Fig. 2. Pterygiophore numbers and abdominal, caudal, and total vertebrae were counted from X-rays taken on a Faxitron UltraFocus X-ray cabinet (settings: 38 kV, 4.8 s). Spine lengths were measured on X-rays using Fiji version 2.0.0[88] and adjusted by taking residuals from multiple regression against standard length and sex. Presence or absence of a fourth spine and Pterygiophore numbers (6 or more than 6) were coded as binary traits (0 or 1). Phenotypes were analysed in R/qtl using Haley–Knott regression via the scanone function, with a normal model for the length traits and a binary model for the spine and pterygiophore number[87]. For the vertebral counts, a non-parametric scanone analysis was done. Permutation tests ($n = 1,000$) were used to establish the logarithm of the odds (LOD) significance thresholds ($\alpha = 0.05$) for each trait. The analysis is based on 340 F2 fish and a set of 452 SNP markers.

**In situ hybridization probes.** RNA was extracted by homogenizing 10–20 stage 19/20 embryos in TRIzol using a FastPrep-24 machine (MP Biomedicals) and lysing matrix M. RNA was washed once with chloroform, precipitated with isopropanol and resuspended in diethyl pyrocarbonate water. RNA was treated with on-column DNase and was cleaned up using the QIAGEN RNeasy Mini (catalogue no. 74104). Complementary DNA (cDNA) was made with SuperScript VILO cDNA Synthesis (catalogue no. 11754050; Thermo Fisher Scientific). For each riboprobe, PCR with reverse transcription amplification was done with the primers shown in Supplementary Table 2. The *HOXD11B* probe was cloned into pCR2.1-TOPO (catalogue no. K450001; Invitrogen) in both orientations; the *HOXD4B* and *HOXD9B* probes were cloned into pCRII-Blunt II-TOPO (catalogue no. K280020; Invitrogen) in both orientations. The vectors containing the probe sequences were linearized with BamHI (catalogue no. FD0054; Thermo Fisher Scientific), and the sense and antisense probes were in vitro transcribed with T7 RNA Polymerase (catalogue no. P2075; Promega Corporation).

**Whole-mount in situ hybridization.** Whole-mount in situ hybridizations at Swarup stages 19–20 were done as described by Thisse & Thisse[89] with the following modifications. Embryos were manually dechorionated with forceps (catalogue no. 11251-10; Fine Science Tools) after overnight fixation in 4% paraformaldehyde in PBS. To remove the pigmentation, they were bleached for 10 min in 0.8% KOH, 3% hydrogen peroxide (30%) and 0.1% Tween 20. Finally, embryos were permeabilized with proteinase K for 10 s at 10 µg ml$^{-1}$ in PBS with 0.1% Tween 20.

**GFP knock-in.** CRISPR–Cas9 was used to generate GFP reporter lines for *HOXD11B*, as described elsewhere[90]. Cas9 protein (QB3 MacroLab, University of California at Berkeley), a donor plasmid (pTia1l-hspGFP, deposited at Addgene, containing hsp70, GFP and a single guide RNA (sgRNA) target site), and two sgRNAs were injected. One sgRNA (*HOXD11B*-GFP-sgRNA; Supplementary Table 3) targeted the region 346 bp upstream of the endogenous *HOXD11B* start codon, and one targeted the donor plasmid. The *HOXD11B*-GFP-sgRNA was designed as described previously[91]. Tia1l sgRNA was used to cut the plasmid[92] and has a sequence not present in the *Gasterosteus aculeatus* genome. The injection mix contained a final concentration of 1 µg µl$^{-1}$ Cas9 protein, 31 ng µl$^{-1}$ Tia1l sgRNA, 31 ng µl$^{-1}$ *HOXD11B*-GFP-sgRNA, and 0.05% phenol red; it was adjusted to the final concentration with 10 mM Tris, pH 7.5.

Fertilized eggs from *Gasterosteus* LITC fish were injected at the single-cell stage and embryos were screened at stage 20 (approximately 84 hours post-fertilization) for GFP expression. GFP$^+$ fish were imaged again at stages 29–31 (18 days post-fertilization). Fry were anaesthetized with 3 mg$^{-1}$ tricaine. Imaging was done with an MZ FLIII fluorescence stereomicroscope (Leica Microsystems) using GFP2 filters and a ProgResCF camera (Jenoptik). GFP$^+$ fish were grown to adulthood and crossed to wild-type LITC fish. Progeny were screened at stage 20 for GFP expression. To confirm integration and orientation of the GFP construct, primers were designed upstream and downstream of the *HOXD11B*-GFP-sgRNA site and in the plasmid on either side of the sgRNA cut site within the hsp70 promoter or the TOPO backbone (Supplementary Table 2). All combinations of primers were tested by PCR. The presence and absence of bands were used to determine orientation. Sanger sequencing was used to determine exact integration sites.

**HOXD11B coding and regulatory mutations using CRISPR–Cas9.** Mutations in the *HOXD11B* coding regions were generated by injecting Cas9 protein and an sgRNA targeting the first exon (*HOXD11B*-coding-sgRNA; Supplementary Table 3). The sgRNA was designed and synthesized as described previously[91] and injected at 300 ng µl$^{-1}$ with 1 µg µl$^{-1}$ of Cas9 protein and 0.05% phenol red in 10 mM Tris, pH 7.5, into fertilized eggs from two anadromous *Gasterosteus* populations (LITC and RABS) at the single-cell stage. Mutations were confirmed by PCR (using Phusion High-Fidelity DNA Polymerase (catalogue no. F-530L; Thermo Fisher Scientific), GC buffer and 3% dimethylsulfoxide (DMSO) with *HOXD11B*-coding_1F and 1R; Supplementary Table 2) amplified at 98 °C (3 min), then 35 cycles at 98 °C (10 s)/60 °C (30 s)/ 72 °C (30 s), and a final extension at 72 °C for 10 min.

Two strategies were used to delete *AxE*, the conserved enhancer (466 bp) between *HOXD9B* and *HOXD11B*: (1) 3 sgRNAs (*AxE*-sgRNA_1, 2 and 3;

Supplementary Table 3) and a 60 bp repair phosphorothioate modified oligonucleotide (Integrated DNA Technologies) with 30 bp of homology to either side of the enhancer[93]; or (2) a total of 6 sgRNAs (*AxE*-sgRNA_1 through 6; Supplementary Table 3) targeting the edges and middle of the enhancer. The sequence for the repair oligo was G*A*A CGT AAA AGG ATT CAG GAG CTC AAG CGA GTC GGT TCC AAA CGT GTC GTT GCC CAG C*A*G (the asterisks indicate phosphorothioate bonds). If the first two bases of the sgRNA target sequences were not Gs, then they were replaced to aid in the transcription of the sgRNA. The injection mix included 1 µg µl⁻¹ of Cas9 protein, 300 ng µl⁻¹ of total of the sgRNAs (100 ng µl⁻¹ of each for strategy 1, 50 ng µl⁻¹ of each for strategy 2), 1.5 pmol µl⁻¹ repair oligonucleotide (strategy 2 only), 300 mM KCl[94], 0.05% phenol red. Mutations were confirmed by PCR as described above, except that the extension time was 1 min, and the annealing temperature was 64 °C. Two sets of primers were used, with the first amplifying only the 571 bp region including the enhancer and the second including approximately 3.6 kb around the enhancer (Supplementary Table 2).

***Apeltes quadracus* association mapping.** *Apeltes quadracus* were collected in May 2018 and May and July 2019 using minnow traps and dip nets from Fortress Louisbourg (site 325) and Tidnish River Site 3 (site 171)[50] (GPS coordinates in Supplementary Table 1). Sticklebacks were euthanized as described above and fixed in 70% ethanol or Alfred Lamb's Navy Dark Rum 151 Proof. *Apeltes* anatomical traits and landmarks are diagrammed in Extended Data Fig. 7. Fish were phenotyped for spine number by X-ray as described above. The spine lengths and standard lengths of Louisbourg fish were measured in triplicate using digital callipers, averaged, and used to calculate residuals for each spine against the standard length of the fish.

To identify *Apeltes* genotyping markers, *HOXDB* sequences were amplified by PCR using primers (*PUNG-GAC*_1-11; Supplementary Table 2) conserved between the *Gasteosteus aculeatus*[43] and *Pungitius pungitius* genomes (GenBank assembly accession nos. GCA_003399555.1, GCA_003935095.1, GCA_902500615.2)[95,96] (Supplementary Table 2). PCR products were cloned into pCRII-Blunt II-TOPO, miniprepped and Sanger-sequenced from 2–4 individuals with differing spine numbers to identify variable regions. Additional regions were filled by designing primers spanning the initial products. These PCR products were also cloned and Sanger-sequenced using primers shown in Supplementary Table 2.

Twelve markers were identified throughout the *Apeltes HOXDB* cluster and scored in 211 fish from Louisbourg Fortress (7 six-spine, 99 five-spine, 104 four-spine, 1 three-spine) and 121 fish from Tidnish River 3 (1 six-spine, 59 five-spine, 59 four-spine, 1 three-spine, 1 two-spine).

Microsatellite markers were amplified using the universal fluorescent primer system[97]. A 20 µl PCR reaction mixture contained 2× Master Mix (catalogue no. K0171; Thermo Fisher Scientific), 0.5 µM 6FAM M13 forward universal primer, 0.125 µM forward primer, 0.5 µM reverse primer and 10 ng genomic DNA. The PCR programme was 94 °C (5 min), 30 cycles at 94 °C (30 s)/58 °C (45 s)/72 °C (45 s), 8 cycles at 94 °C (30 s)/53 °C (45 s)/72 °C (45 s) and a final extension at 72 °C for 10 min. For *AQ-HOXDB*_2, the cycle number was reduced from 30 to 27. PCR was cleaned using ExoSAP-IT PCR Product Cleanup Reagent (catalogue no. 78205.1.ML; Applied Biosystems), fragment sizes were analysed on an Applied Biosystems 3730xl Genetic Analyzer, and peaks were called using the Microsatellite plugin for Geneious version 10.2.3.

Indel and SNP markers were scored using PCR with 2× Master Mix, 0.5 µM forward primer, 0.5 µM reverse primer and 10 ng genomic DNA. The PCR programme was 95 °C (5 min), 35 cycles at 95 °C (30 s)/54 °C (45 s)/72 °C (30 s) and a final extension at 72 °C for 10 min. The one exception was *AQ-HOXDB*_6, where PCR was done with Phusion High-Fidelity DNA Polymerase, GC buffer and 3% DMSO; the PCR programme was 98 °C (3 min), 35 cycles at 98 °C (10 s)/60 °C (30 s)/72 °C (10 s) and a final extension at 72 °C for 10 min. The *AQ-HOXDB*_5 PCR product was digested with BssSI-v2 (catalogue no. R0680L; New England Biolabs); the *AQ-HOXDB*_6 PCR product was digested with NdeI (catalogue no. FD0583; New England Biolabs). PCR products were run on a 2% agarose gel to score size differences.

Allele frequencies in low-spine (two- to four-spine) and high-spine (five- to six-spine) fish were compared using CLUMP version 2.4[98], which performs a modified chi-squared analysis to determine the significance of allele frequency differences. For microsatellite markers, the negative log *P* values of the chi-squared value (T4) from the 2 × 2 contingency table generated by CLUMP are shown in Fig. 3. For indel or SNP markers, a chi-squared test was performed in R v.3.6.1. Associations between residual spine lengths and genotypes were quantified using an analysis of variance performed in R.

***Apeltes* genome assembly.** Whole-genome sequencing (WGS) using 10X Genomics Chromium-linked read technology was performed on two *Apeltes quadracus* from the Louisbourg Fortress population (one four-spine and one five-spine). Genomic DNA was extracted from brains and prepared using the QIAGEN MagAttract HMW DNA kit. Linked read data from each fish were assembled using Supernova v.2.1.1 with default settings[99]. The four-spine *Apeltes* assembly had 16,216 scaffolds with 416,290,932 bases (scaffold N50: 393,888 bp; L50: 247; N90: 7,174 bp; L90:

3,684). The five-spine *Apeltes* assembly had 24,175 scaffolds with 397,678,333 bases (scaffold N50: 69,128 bp; L50:1,629; N90: 4,805 bp; L90: 10,192).

To use GATK version 4.1.4.1 in the allele-specific RNA-seq pipeline, the genome needed to be on fewer scaffolds than generated by linked read data. To achieve this, we started with the four-spine *Apeltes* assembly and assumed that the chromosome structure of *Apeltes* and *Gasterosteus* are similar. We used a reference-guided scaffold approach by generating global genome to genome alignments with minimap2 (ref. [100]) and MUMmer version 4.0.0[101]. The alignment information was processed by RaGOO version 1.1[102] to order and orient contigs into scaffolds, which resulted in the *Apeltes* genome reference used in the GATK allele-specific RNA-seq pipeline.

**High-spine *Gasterosteus* genome assembly.** WGS using 10X Genomics Chromium-linked read technology was performed on two four-spine *Gasterosteus aculeatus* from the F5 generation of the Boulton-Bodgea Bay QTL cross. Genomic DNA was extracted from the brains of the fish and prepared using the QIAGEN MagAttract HMW DNA kit. The linked read data of each fish were assembled using Supernova v.2.1.1 with default settings[99].

WGS using PacBio HiFi was also performed on one four-spine *Gasterosteus aculeatus* from the F5 generation of the Boulton-Bodega Bay QTL cross. Genomic DNA was extracted from the testes of the fish and prepared using the QIAGEN MagAttract HMW DNA kit. The genome was assembled using Canu version 2.1.1. The purge haplotigs pipeline (https://bitbucket.org/mroachwri/purge_haplotigs/src/master/) was used to phase the alleles and identify the contigs that appeared twice in the assembly. The final assembly had 483 scaffolds with a total of 489,328,730 bases (scaffold N50: 3,689,351 bp; L50: 37; N90: 633,554 bp; L90: 166).

**Transgenic enhancer assays.** To identify and confirm sequence variants in the intergenic region between *HOXD9B* and *HOXD11B*, the approximately 6 kb intergenic region from *Apeltes* was amplified from a three-spine and a six-spine *Apeltes* from the Louisbourg population and a three-spine and a six-spine from the Tidnish population with Phusion High-Fidelity DNA Polymerase in GC buffer and 3% DMSO using the primers in Supplementary Table 2. The resulting products were TOPO-cloned into pCRII-Blunt II-TOPO. Colonies were miniprepped and Sanger-sequenced. To generate the plasmids for the enhancer assay, the low- and high-spine versions of the approximately 600 bp region that contains *AxE* were then amplified with primers[28,103] that included overhangs homologous to the PT2HE GFP reporter vector[28,103]. The reporter vector was cut with EcoRV (catalogue no. ER0201; Thermo Fisher Scientific), and the insert and vector were joined using Gibson Cloning (catalogue no. E2611S; New England Biolabs). The resulting plasmids were screened by SacI (catalogue no. ER1131; Thermo Fisher Scientific) restriction digest and further Sanger-sequenced to check for mutations. The 587 bp high-spine and 611 bp low-spine *Apeltes AxE* sequences are available in GenBank at OK383404 and OK383405, respectively.

To test the enhancer activity of portions of the transposable element insertions found in the high-spine Boulton allele, the LTR from the ERV and LINE were amplified with the primers BOUL-*HOXDB*_LTR and BOUL-*HOXDB*_LINE, respectively. The resulting products were inserted into the PT2HE GFP reporter vector as described above.

Transgenic *Gasterosteus aculeatus* sticklebacks were generated by microinjection at the single-cell stage from LITC *Gasterosteus aculeatus* as described in Chan et al.[34]. Plasmids (25 ng µl⁻¹) were injected with *Tol2* transposase messenger RNA (36 ng µl⁻¹) and 0.1% phenol red as described in Hosemann et al.[104]. *Tol2* mRNA was synthesized by in vitro transcription using the mMessage mMachine SP6 kit (catalogue no. AM1340; Invitrogen) from the pCS-TP plasmid[105] cut with Bsp120I (catalogue no. ER0131; Thermo Fisher Scientific). Transgenics were imaged at stage 20 (approximately 84 hours post-fertilization) and stage 29/31 (approximately 18–30 days post-fertilization) as described in the GFP knock-in section above. The hsp70 promoter drives expression in the lens of the eye by 9 d post-fertilization[106]. At stage 29/31, bilateral lens GFP expression was used to identify less mosaic fish. For all statistics, to compare expression patterns between different constructs, a Fisher's exact test with a Bonferroni-corrected significance threshold at $\alpha = 0.05$ was used.

**dN-dS calculation.** dN-dS calculations were performed in R using ape v.5.3 (ref. [107]). Sequence alignments for each gene (*HOXD11B*, *HOXD9B*, and *HOXD4B*) were generated in Geneious using translation alignment. *Gasterosteus* transcripts were based on splicing patterns validated from cDNA. Sequences for *P. pungitius* were determined by BLASTN version 2.7.1[108] of *Gasterosteus* exons against *Pungitius*: GCA_003935095.1 (ref. [95]). Sequences for *Apeltes* were identified from our genome assembly. Sequences for *Gasterosteus wheatlandi*, *Culaea inconstans* and *Spinachia spinachia* were identified by BLAST version 2.7.1 of *Gasterosteus* exons against unassembled short reads from WGS of the respective species[109] (C. Peichel and M. Hiltbrunner, personal communication).

**RNA-seq.** For *Gasterosteus* RNA-seq, a lab-raised LITC anadromous female with three dorsal spines was crossed to a high-spine *Gasterosteus* male with five dorsal spines. The high-spine *Gasterosteus* line is the F5 generation of the original QTL cross between Boulton and Bodega Bay. The fish were selected for high

spine number; by F5, more than 80% of fish have 4 or more dorsal spines. To confirm that the fish carried the Boulton allele at the *HOXDB* locus, the allele was amplified using BOUL-*HOXDB*_1F and 1R (Supplementary Table 2) with Phusion High-Fidelity DNA Polymerase in GC buffer and 3% DMSO using a 2-step PCR programme (94 °C (1 min), 30 cycles at 98 °C (10 s)/68 °C (15 min) and a final extension at 72 °C for 10 min) and run on an agarose gel. The Boulton allele is approximately 15 kb, and the BDGB allele is approximately 1.9 kb. The sequences of the two alleles are available at OK383406 (BDGB) and OK383407 (Boulton) in GenBank.

The resulting clutch was raised to 11–13 mm. Fry were euthanized as described above and dissected on a 2% agarose plate with size 00 insect pins and Spring Scissors (catalogue no. 15000-08; Fine Science Tools). Spine, fin and pterygiophore tissues (Fig. 2a) were collected and flash-frozen in liquid nitrogen in FastPrep Tubes (catalogue no. MP115076200; MP Biomedicals). DNA from tails was amplified and genotyped to ensure fish had informative SNPs in the coding regions of *HOXD11B* and *HOXD9B* using *HOXD11B*-coding_1F and 1R primers, *HOXD9B*-coding_1F and 1R primers (Supplementary Table 2) and the PCR conditions described above.

Based on genotyping, 12 three-spine progeny and 6 four-spine progeny were chosen for RNA extraction, library preparation and sequencing. Samples for RNA extraction were homogenized using MP FastPrep 2 ×20 s with Matrix M with a 5 min rest in between. RNA extractions were performed using NucleoSpin RNA XS (Takara Bio) and resuspended in 20 µl RNase-free water. RNA was quantified by Qubit (Invitrogen) using the HS Assay Kit (catalogue no. Q32851; Invitrogen). A subset of samples was quality-controlled to check the RNA integrity number (RIN) values by BioAnalyzer using the RNA 6000 Pico Kit (catalogue no. 5067-1513; Agilent Technologies). The RINs were between 8.2 and 10, with most higher than 9.6. Sequencing libraries were generated with the Illumina Stranded mRNA Prep kit (catalogue no. 20040532) and 20–100 ng RNA. PCR cycle numbers were determined by quantitative PCR (generally 12 cycles for embryo samples with 200 ng RNA, 14 cycles for dorsal spine and pterygiophore samples with 100 ng RNA, 13 cycles for dorsal and anal fin samples with 100 ng RNA and 15 cycles for samples with less than 100 ng RNA input). Quality control of libraries was done by Qubit with a dsDNA HS Assay Kit to check concentrations and by BioAnalyzer with a high-sensitivity kit (catalogue no. 5067-4626; Agilent Technologies) to check sizes. Libraries were sequenced to approximately 30 million read coverage on a NovaSeq 6000 (2 ×150 bp) by NovoGene. Reads were trimmed with Cutadapt version 2.4[110] using the TrimGalore version 0.6.6 wrapper (https://github.com/FelixKrueger/TrimGalore) and mapped to the *gasAcu1-4* reference genome (https://doi.org/10.5061/dryad.547d7wm6t) with STAR version 2.7.10a two-pass mapping[111]. For allele-specific expression analysis, base quality was adjusted and variants were called using GATK version 4.1.4.1 as recommended by the Broad Institute[112,113]. Reads at each site were counted using GATK ASEReadCounter version 4.1.4.1. We required that SNPs be called as heterozygous in at least 1 tissue of each fish, that the number of reads at a given site be greater than 12 (three-spine) or 14 (four-spine) for each fish and that the overall minor allele frequency be greater than 5%. To quantify allele-specific expression differences seen between the dorsal tissues and anal fin (control), we took log₂ ratios of reference reads to alternate reads in each sample and compared each dorsal tissue to the anal fin using a Mann–Whitney *U*-test.

To improve gene predictions and recover new transcripts for differential gene expression analysis, StringTie version 2.1.4 was used along with the existing Ensembl annotations[114]. Given the large number of reads, BAM files were filtered by quality, downsampled to 20% and merged into one file that was used as the input for StringTie. The merge function was used to add genes from the Ensembl annotations not present in the sequenced samples. All *Hox* genes were manually checked. In some cases, the two genes were merged into one due to their close proximity; these were manually separated in the GTF file. FeatureCounts version 1.6.0 was then used with the new GTF file to assign reads to genes[115]. Differential gene expression between different tissues was performed in DESeq2 version 1.26.0 (ref. [116]).

*Apeltes* RNA-seq followed similar protocols as *Gasterosteus*, with the following differences. Spines, blank pterygiophores, dorsal fin and anal fin were dissected from Louisbourg *Apeltes* clutches raised in the lab to 11–13 mm. The fry were genotyped for the 2 peak association mapping markers (AQ-*HOXDB*_6 and 7). For allele-specific expression analysis, four fish with the L/LHR genotype, four fish with the H/L genotype and three fish with the H/LHR genotype were sequenced. Three four-spine L/L genotypes were also sequenced to examine expression differences between tissues. To generate gene predictions for the *Apeltes* genome, StringTie was used; the *Hox* genes were identified by BLAST[108] and manually named in the GTF file. For high-spine fish, allele-specific analysis was performed as described above for *Gasterosteus*. The *Apeltes* 10X-linked read data were used as input for 10X Long Ranger v.2.2.2 to generate a VCF file of known variants for GATK BaseRecalibrator version 4.1.4.1. Because the clutch size of *Apeltes* is smaller and thus the number of replicates was lower than for *Gasterosteus*, we used a two-tailed Fisher's exact test to compare expression in dorsal tissues to the anal fin. We summed the references and alternate reads within each tissue to generate a 2×2 contingency table. For the analysis shown in Extended Data Fig. 5, differential gene expression between tissues was performed in DESeq2 (ref. [116]).

**Reporting summary.** Further information on research design is available in the Nature Research Reporting Summary linked to this article.

## Data availability
Raw and processed allele-specific RNA-seq data are available in the National Center for Biotechnology Information (NCBI) Gene Expression Omnibus database under accession no. GSE184888 (subseries GSE184885, GSE184886, GSE184887, GSE190498). PacBio HiFi and 10X-linked read data from *Gasterosteus* high-spine sequencing are available at the NCBI under BioProject no. PRJNA766710. The 10X-linked read data from *Apeltes quadracus* four- and five-spine fish are available under BioProject no. PRJNA769115. Sequences surrounding *AxE* in *Gasterosteus* from the two parental QTL populations and the *Apeltes AxE* sequences tested in the transgenic assays are available in GenBank (OK383406, OK383407, OK383404, OK383405). QTL mapping files, phenotype data files and association mapping genotype files are available at Figshare (https://doi.org/10.6084/m9.figshare.20033063). The pTia1l-hspGFP plasmid is available from Addgene. Source data are provided with this paper. Other materials will be made available upon reasonable request.

## Code availability
The code is available at Figshare (https://doi.org/10.6084/m9.figshare.20033063).

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

## Acknowledgements

We thank C. Peichel, M. Fuller, W. Talbot, R. Nusse, E. Kingsley, H. Hoekstra and members of the Kingsley lab for helpful discussions. We thank the Semiahmoo First Nation for allowing us to collect sticklebacks from the Little Campbell River (British Columbia, Canada). We thank B. Summers for setting up the F0 QTL cross, D. Desmet for initial measurements of some of the QTL cross fish, H. Folse for initial testing of CRISPR GFP knock-in, A. Thompson, K. Thompson and D. Schluter for help collecting sticklebacks from the Little Campbell River, and C. Peichel and M. Hiltbrunner for *G. wheatlandi*, *C. inconstans* and *S. spinachia* HOXD11B sequences. This work was supported in part by NIH graduate training grant no. 2T32GM007790 (J.I.W.), predoctoral fellowship from the National Science Foundation (T.R.H., G.A.R.K.), a Stanford Graduate Fellowship (G.A.R.K.), a Helen Hay Whitney Postdoctoral Fellowship (A.L.H.), NIH grant no. R01GM124330 (M.A.B.) and Natural Sciences and Engineering Research Council of Canada Discovery Grant no. RGPIN-2016-04303 (A.C.D.). D.M.K. is an investigator of the Howard Hughes Medical Institute.

## Author contributions

J.I.W., T.R.H. and D.M.K. conceptualized the study. J.I.W. and T.R.H. carried out the formal analysis. J.I.W., T.R.H., J.N.A., E.H.A., G.A.R.K., S.D.B. and A.L.H. carried out the investigation. T.E.R., M.A.B., C.B.L., A.C.D. and D.M.K. managed the resources. J.I.W. and T.R.H. curated the data. J.I.W. and D.M.K. wrote the original manuscript. J.I.W., T.R.H., J.N.A., E.H.A., G.A.R.K., S.D.B., A.L.H., T.E.R., M.A.B., C.B.L., A.C.D. and D.M.K. reviewed and edited the manuscript. J.I.W. visualized the data. D.M.K. supervised the study. C.B.L., A.C.D. and D.M.K. acquired the funding.

## Competing interests

The authors declare no competing interests.

## Additional information

**Extended data** is available for this paper at https://doi.org/10.1038/s41559-022-01855-3.

**Correspondence and requests for materials** should be addressed to David M. Kingsley.

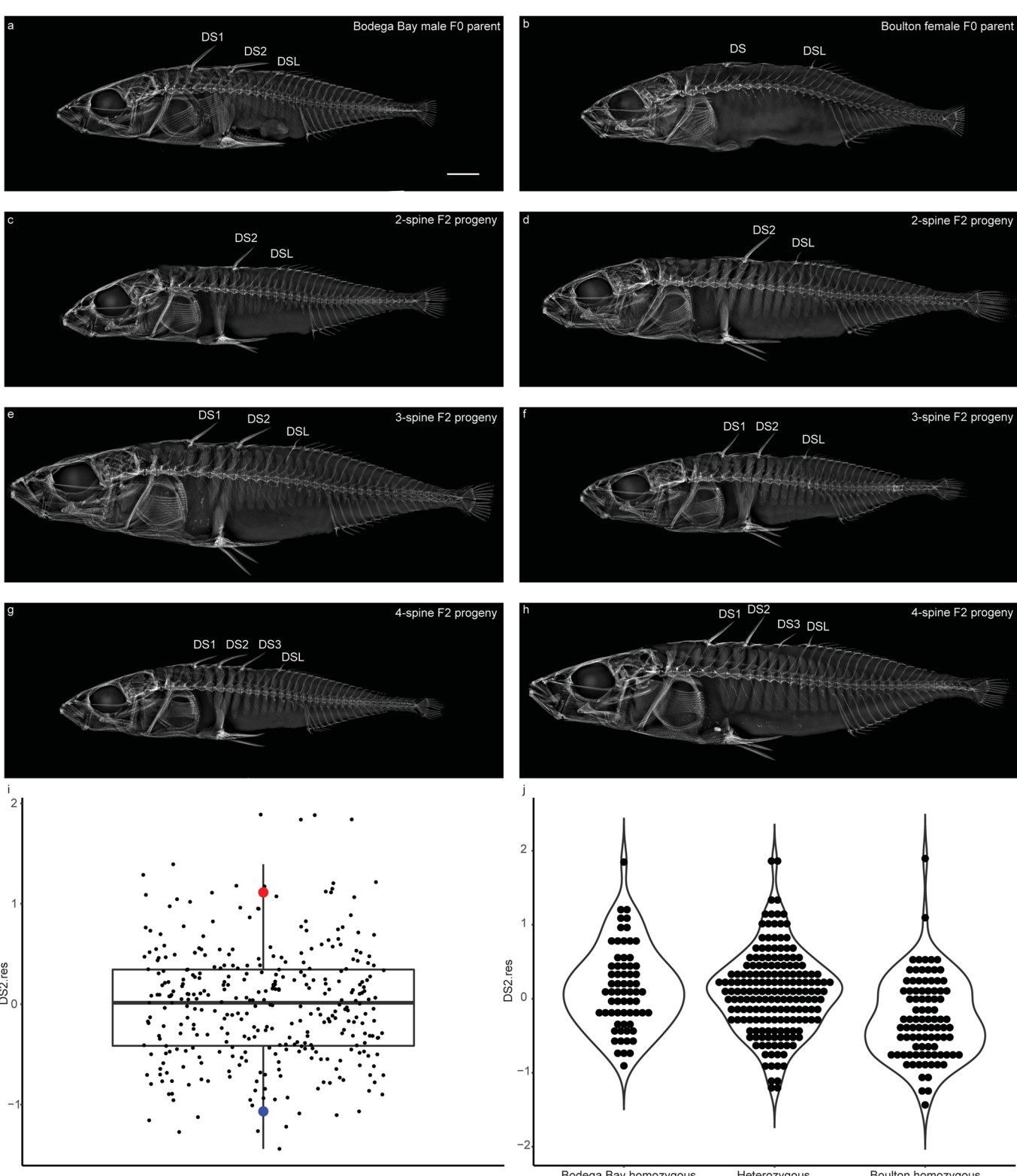

**Extended Data Fig. 1 | Spine phenotypic variation in *Gasterosteus* populations and F2 hybrids. a**. Bodega Bay male parent and **b**. Boulton Lake female parent of the QTL cross. Note that the anterior-most dorsal spine in Boulton is located at a position intermediate between the first two spines in typical *Gasterosteus*[39] and is here labeled DS because it cannot be unambiguously assigned as DS1 or DS2. Representative F2 progeny showing a range of spine numbers, including: **c**. and **d**., two dorsal spines; **e**. and **f**., three dorsal spines; **g**. and **h**., four dorsal spines. Scale bar is 5 mm. **i**. The distribution of DS2 residual lengths (DS2.res) in F2 progeny of the QTL cross (n = 340). In the box and whisker plot: center line, median; box limits, interquartile range; whiskers, 1.5x interquartile range; each measurement is represented by a single point. Red and blue points are included for DS2 and DS of the Bodega Bay and Boulton parents of the cross, respectively. The residual was calculated with respect to standard length and sex. **j**. Distribution of DS2.res as a function of genotype at the peak marker on the distal end of chr6 (Fig. 1b).

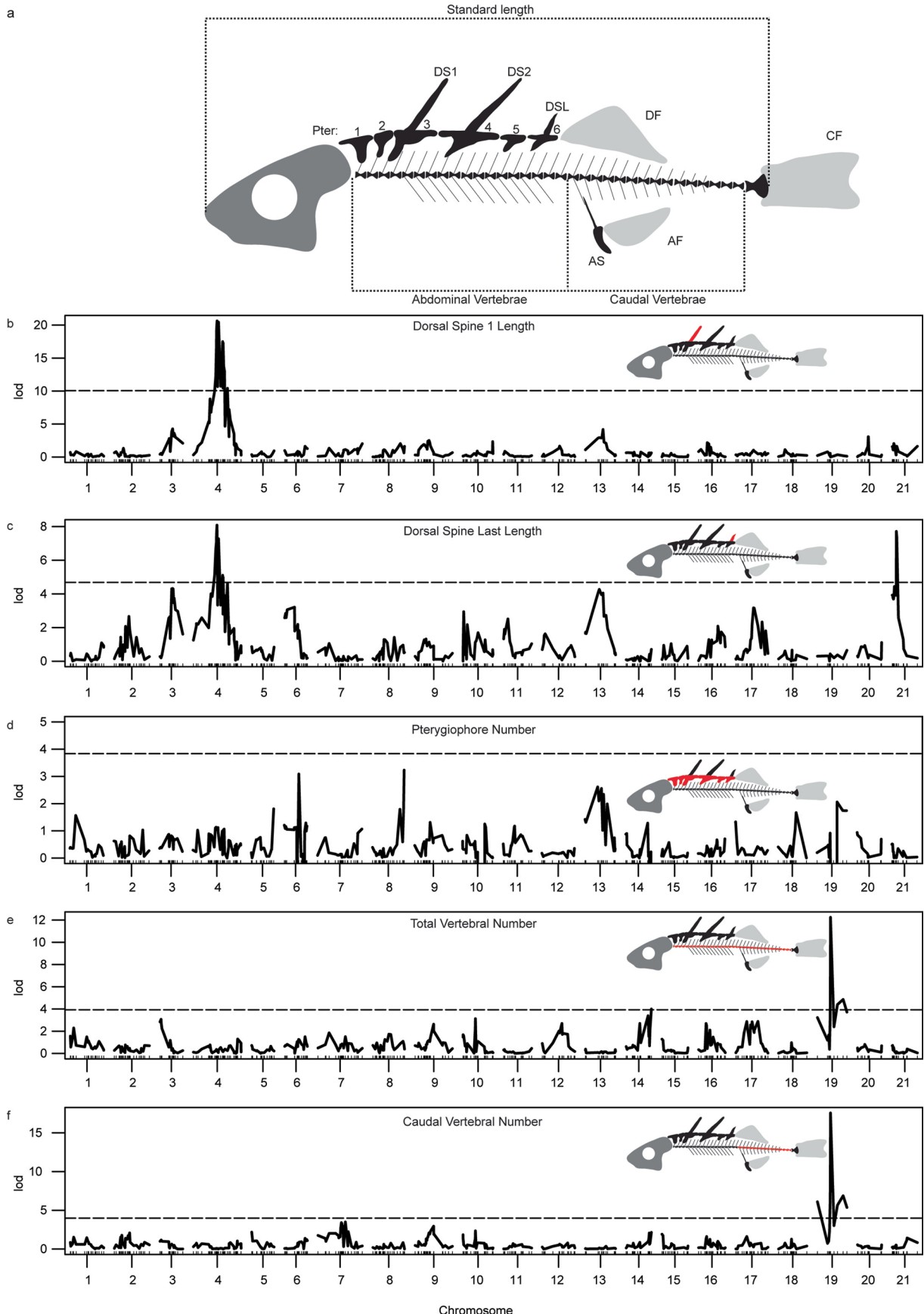

**Extended Data Fig. 2 | See next page for caption.**

**Extended Data Fig. 2 | QTL mapping of other spine lengths and axial traits. a**. Schematic of *Gasterosteus* anatomical features. Most *Gasterosteus* have three dorsal spines that in this study are referred to as dorsal spine 1 (DS1), dorsal spine 2 (DS2), and dorsal spine last (DSL). The dorsal side of the fish also has median bony plates known as pterygiophores, some of which underlie dorsal spines. Typical A-P midline pattern: two non-spine bearing/blank pterygiophores (Pter1 and Pter 2), dorsal spine 1 on pterygiophore 3 (Pter3), dorsal spine 2 on pterygiophore 4 (Pter4), non-spine bearing pterygiophore 5 (Pter5), and dorsal spine last on pterygiophore 6 (Pter6). The three unpaired fins are shown in light gray: dorsal fin (DF), caudal fin (CF), and anal fin (AF). The anal spine (AS) is also indicated on the ventral side of the fish. The standard length shown with the dotted line is from the anterior tip of the jaw to the posterior of the hypural plates. **b**. QTL plot of DS1 length **c**. QTL plot of DSL length **d**. QTL plot of pterygiophore number **e**. QTL plot of total vertebral number **f**. QTL plot of caudal vertebral number. Dotted lines represent genome-wide significance thresholds. Abdominal vertebral number and anal spine length were also tested, but they did not result in any peaks that passed the genome wide significance threshold. The significance threshold (dashed line) is based on LOD scores obtained in 1,000 permutations of the phenotype data ($\alpha = 0.05$).

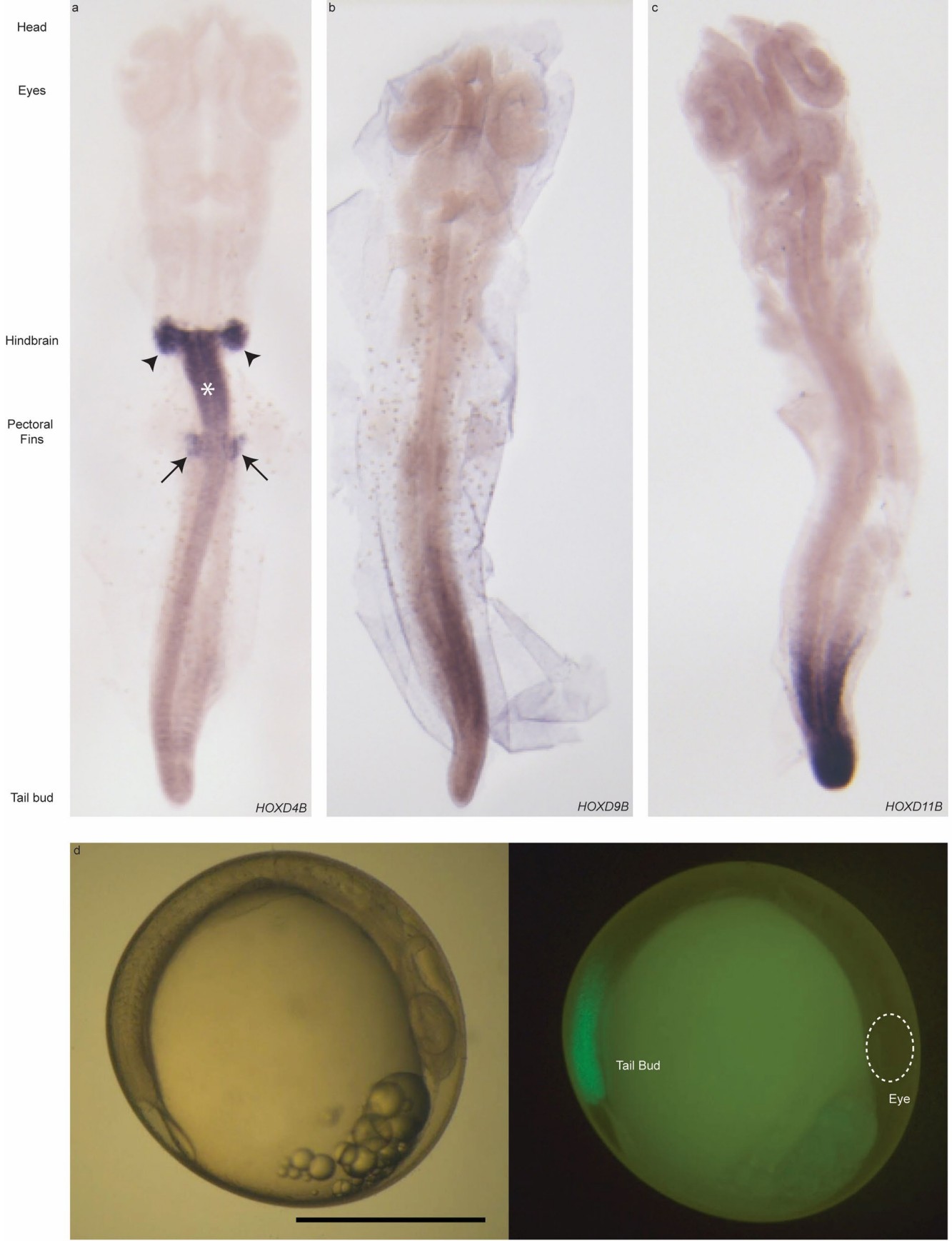

**Extended Data Fig. 3 | See next page for caption.**

**Extended Data Fig. 3 | Embryonic expression of *Gasterosteus HOXDB* genes.** *In situ* hybridization of *Gasterosteus aculeatus* embryos at Swarup stage 19/20 **a**. *HOXD4B* probe. The arrowheads point to the hindbrain, the asterisk indicates the neural tube, and the arrows point to the anterior somites; **b**. *HOXD9B* probe; **c**. *HOXD11B* probe. **d**. eGFP expression at embryonic Swarup stage 19–20 in the tailbud somites from the reporter gene integrated upstream of *HOXD11B* in low-spine *Gasterosteus* (Fig. 1). The eGFP pattern recapitulates the *in situ* hybridization results for *HOXD11B* in panel C. Dotted circle indicates the location of the eye. Scale bar is 1mm.

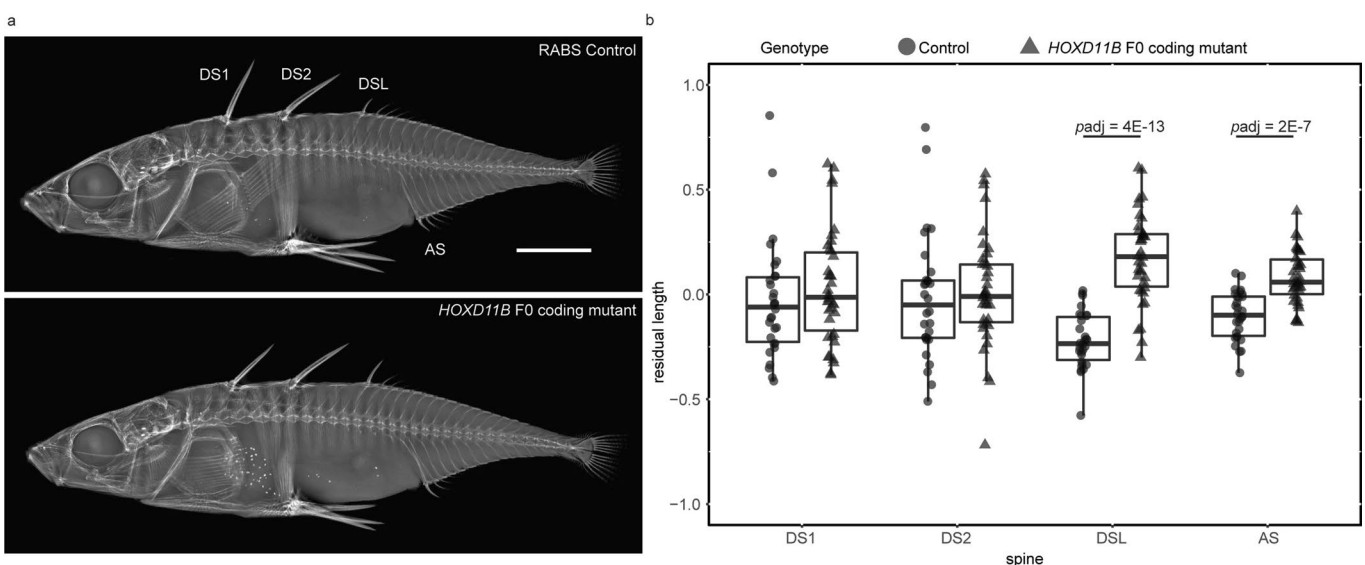

**Extended Data Fig. 4 | Coding mutations in *HOXD11B* cause length changes in stickleback spines in a second anadromous *Gasterosteus* population. a**.
X-rays of an uninjected sibling control RABS *Gasterosteus* (top) and a RABS *Gasterosteus* that was injected at the single cell stage with Cas9 and an sgRNA targeting the coding region of *HOXD11B* (bottom). Scale bar is 5 mm. **b**. Quantification of spine length changes. In the box and whisker plot: center line, median; box limits, interquartile range; whiskers, 1.5x interquartile range; each measurement is represented by a single point (circle for wild type and triangle for mutant). DS1 and DS2 were not significantly different between controls and *HOXD11B* mutant fish. DSL and AS were significantly longer in the F0 mutants compared to the controls (two-tailed t-test Bonferroni-corrected at $\alpha = 0.05$, DSL $p$adj = 4E-13, AS $p$adj = 2E-07, n = 38 injected and n = 30 control from 3 clutches combined). The y-axis is the residual after accounting for the standard length of fish.

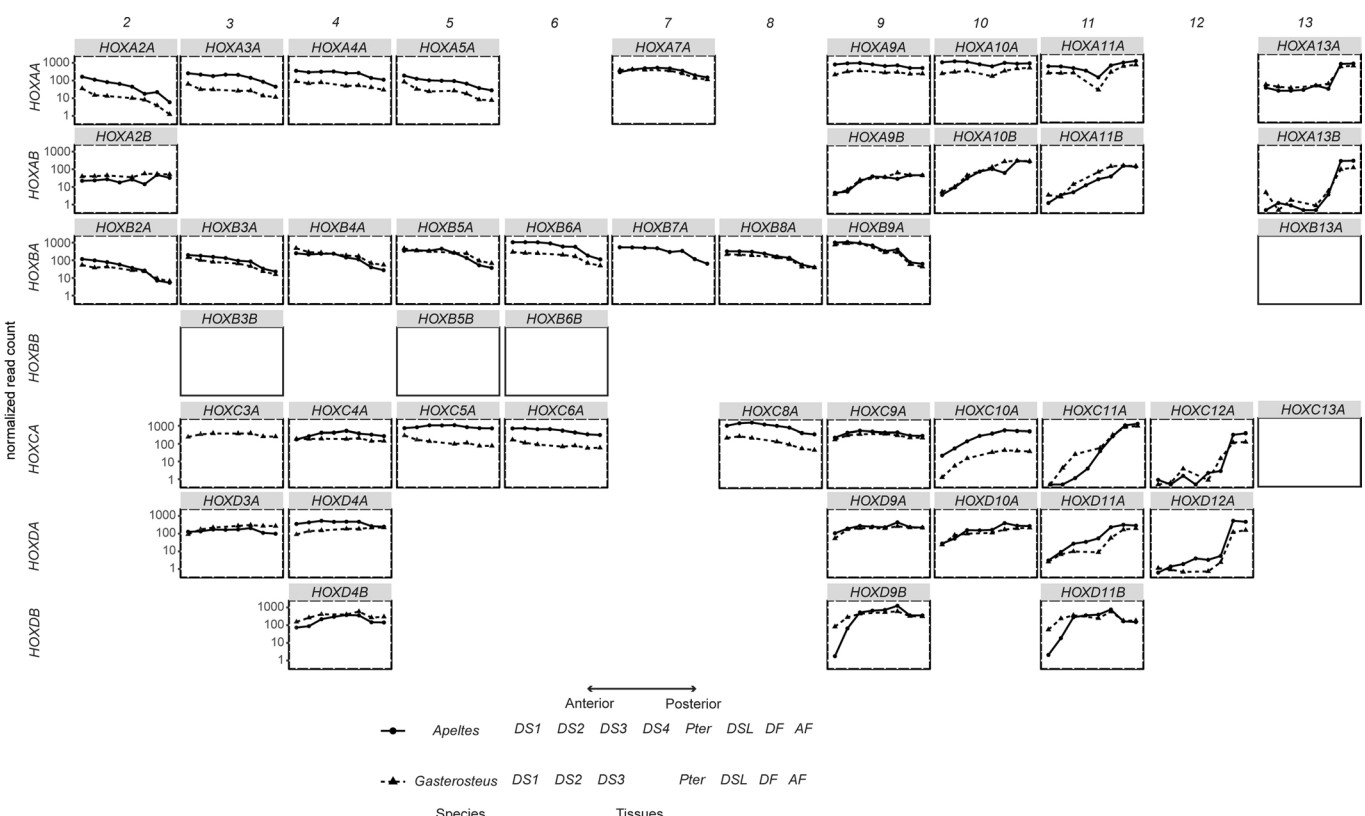

**Extended Data Fig. 5 | *Hox* gene expression patterns in *Gasterosteus* and *Apeltes* spines and fins.** The expression patterns for each *Hox* gene in different stickleback *Hox* clusters are shown with normalized read count on the y-axis and tissue site on the x-axis. The tissues are organized by position from anterior to posterior along the dorsal side of the fish, with the anal fin at the end. The read count shown is the average across all samples for that species; the reads are normalized within each species but not between species. The genes with empty plots exist in both species but are not expressed in the tissues shown with the exception of *HOXB6B* and *HOXB7A*, which are located in gaps in the *Gasterosteus* assembly and thus were not scored; *HOXB6B* is not expressed in *Apeltes*, but *HOXB7A* is expressed. *HOXA1A*, *HOXB1B*, and *HOXB1B* are present in the genomes but were not expressed and are not shown. The genes differentially expressed (*p*adj < 0.01) between the largest anterior spines (DS1 and DS2) in *Gasterosteus* low-spine (three-spine) fish are *HOXA2A*, *HOXA5A*, *HOXA10B*, *HOXC3A*, *HOXC5A*, *HOXC9A*, *HOXC10A*, *HOXC11A*, *HOXD3A*, *HOXD4A*, *HOXD9A*, *HOXD10A*, *HOXD4B*, *HOXD9B*, and *HOXD11B*; the genes differentially expressed (*p*adj < 0.01) between DS1 and DS3 in *Apeltes* low-spine (four-spine) fish are *HOXA10B*, *HOXC4A*, *HOXC8A*, *HOXC9A*, *HOXC10A*, *HOXD9A*, *HOXD10A*, *HOXD11A*, *HOXD4B*, *HOXD9B*, and *HOXD11B*.

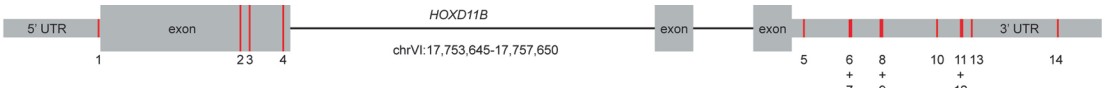

| SNV Number | Position | Heterozygous in cross #1 (embryos) | Heterozygous in cross #2 (spines/fins) |
|---|---|---|---|
| D11B-1 | chrVI:17753999 | Yes | |
| D11B-2 | chrVI:17754511 | | Yes |
| D11B-3 | chrVI:17754547 | | Yes |
| D11B-4 | chrVI:17754674 | Yes | Yes |
| D11B-5 | chrVI:17756571 | Yes | Yes |
| D11B-6 | chrVI:17756735 | | Yes |
| D11B-7 | chrVI:17756750 | | Yes |
| D11B-8 | chrVI:17756854 | Yes | Yes |
| D11B-9 | chrVI:17756868 | Yes | |
| D11B-10 | chrVI:17757061 | | Yes |
| D11B-11 | chrVI:17757143 | Yes | Yes |
| D11B-12 | chrVI:17757146 | Yes | |
| D11B-13 | chrVI:17757191 | Yes | |
| D11B-14 | chrVI:17757502 | | Yes |

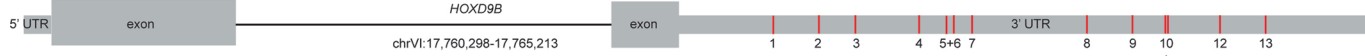

| SNV Number | Position | Heterozygous in cross #1 (embryos) | Heterozygous in cross #2 (spines/fins) |
|---|---|---|---|
| D9B-1 | chrVI:17763039 | Yes | |
| D9B-2 | chrVI:17763202 | Yes | |
| D9B-3 | chrVI:17763337 | Yes | Yes |
| D9B-4 | chrVI:17763561 | | Yes |
| D9B-5 | chrVI:17763661 | | Yes |
| D9B-6 | chrVI:17763688 | | Yes |
| D9B-7 | chrVI:17763754 | | Yes |
| D9B-8 | chrVI:17764176 | | Yes |
| D9B-9 | chrVI:17764343 | Yes | |
| D9B-10 | chrVI:17764466 | | Yes |
| D9B-11 | chrVI:17764474 | | Yes |
| D9B-12 | chrVI:17764664 | Yes | Yes |
| D9B-13 | chrVI:17764838 | | Yes |

| SNV Number | Position | Heterozygous in cross #1 (embryos) | Heterozygous in cross #2 (spines/fins) |
|---|---|---|---|
| D4B-1 | chrVI:17782714 | | Yes |
| D4B-2 | chrVI:17782720 | | Yes |
| D4B-3 | chrVI:17782723 | | Yes |
| D4B-4 | chrVI:17782726 | | Yes |
| D4B-5 | chrVI:17782741 | Yes | |
| D4B-6 | chrVI:17782858 | Yes | |
| D4B-7 | chrVI:17782932 | | Yes |
| D4B-8 | chrVI:17782940 | Yes | Yes |
| D4B-9 | chrVI:17783250 | Yes | |
| D4B-10 | chrVI:17783616 | Yes | Yes |
| D4B-11 | chrVI:17783618 | Yes | Yes |

**Extended Data Fig. 6 | *HOXDB Gasterosteus* nucleotide variants used for allele-specific expression analysis.** With the exception of *HOXD11B*, variants were located in the 3'UTR of the genes. Their locations are indicated by red lines and they are numbered from 5' to 3'. Note that two different crosses were done to generate the embryo and spine/fin samples and because the populations are not inbred the informative variants were not the same. The table below each gene diagram shows which SNVs were informative in which set of samples.

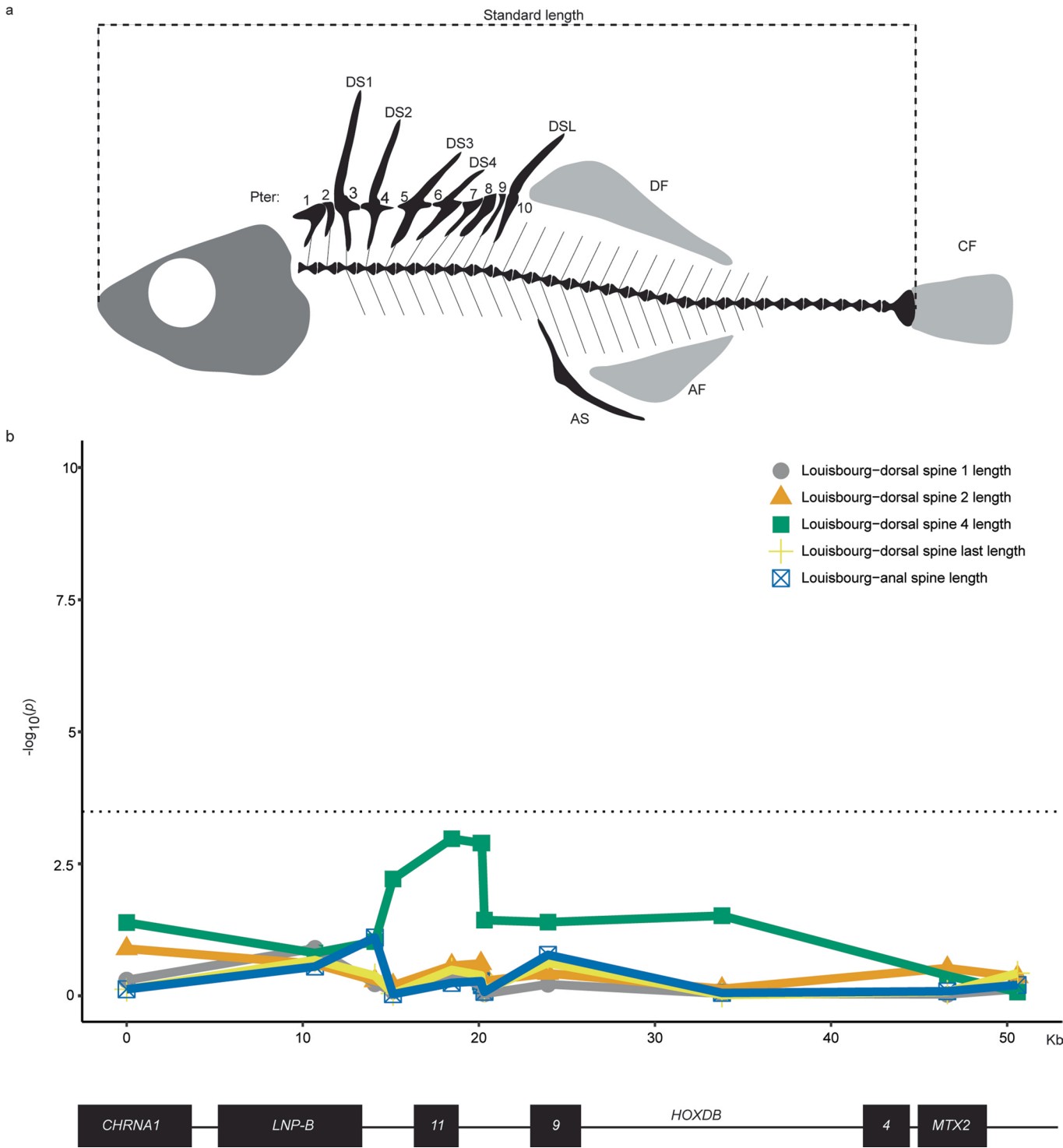

**Extended Data Fig. 7 | Spine anatomy and trait association mapping in Louisbourg *Apeltes*. a**. Schematic of anatomical structures in an *Apeltes* fish with five dorsal spines. Typical A-P midline pattern: two non-spine bearing pterygiophores (Pter 1 and 2), dorsal spine 1 (DS1) on pterygiophore 3 (Pter3), dorsal spine 2 (DS2) on pterygiophore 4 (Pter4), dorsal spine 3 (DS3) on pterygiophore 5 (Pter5), dorsal spine 4 (DS4) on pterygiophore 6 (Pter6), three non-spine bearing pterygiophores (Pter7–9), and dorsal spine last (DSL) on pterygiophore 10 (Pter10). The three unpaired fins are shown in light gray: dorsal fin (DF), caudal fin (CF), and anal fin (AF). The anal spine (AS) is indicated on the ventral side of the fish. The standard length shown with the dotted line is from the anterior tip of the jaw to the posterior of the hypural plates. **b**. The association between *HOXDB* genotypes and length of DS1, DS2, DS4, DSL, and AS were not statistically significant. For significant results with DS3 length, see Fig. 3.

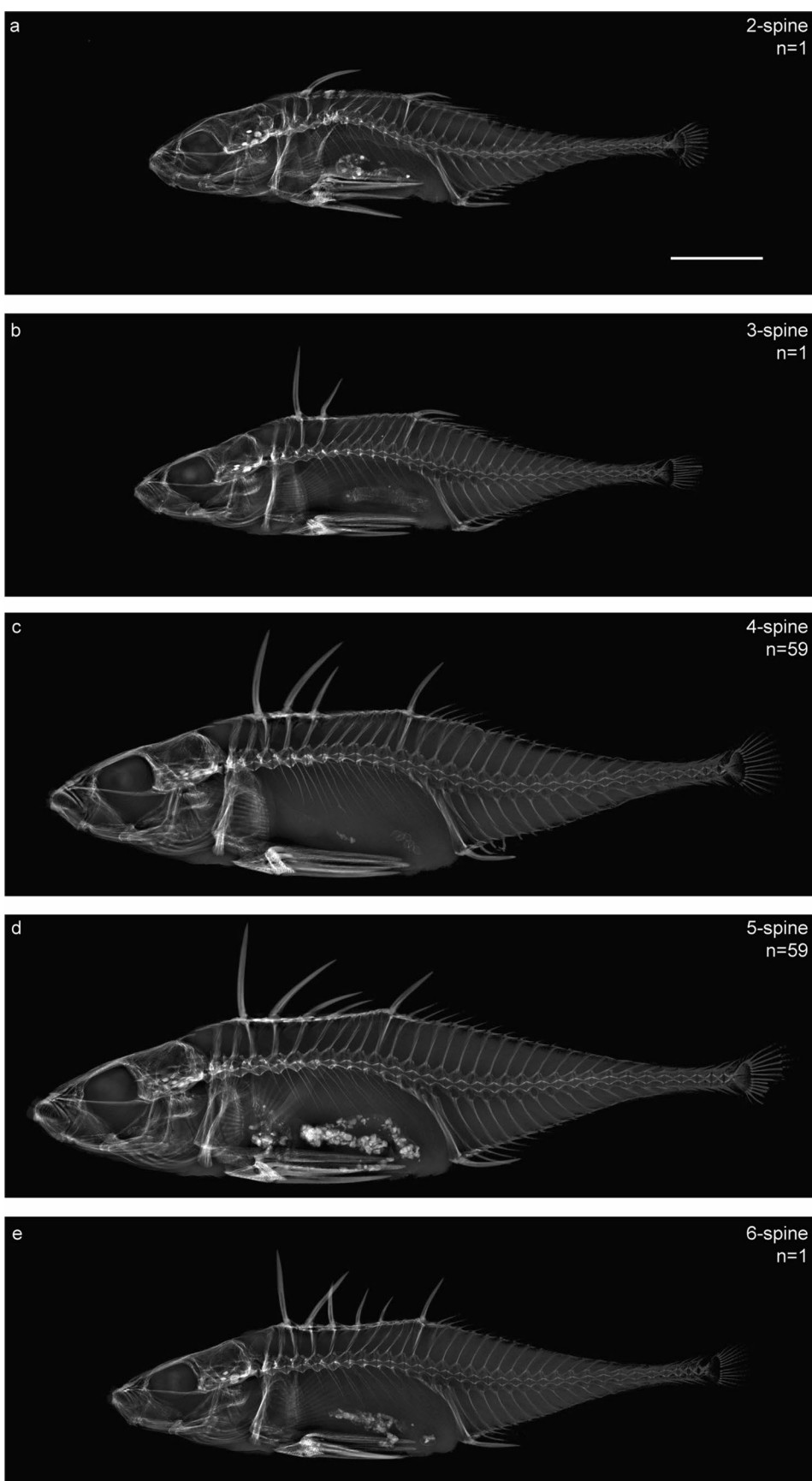

**Extended Data Fig. 8 | Spine number variation in Tidnish *Apeltes*.** X-ray of **a**. two-spine, **b**. three-spine, **c**. four-spine, **d**. five-spine, and **e**. six-spine fish. Scale bar is 5 mm.

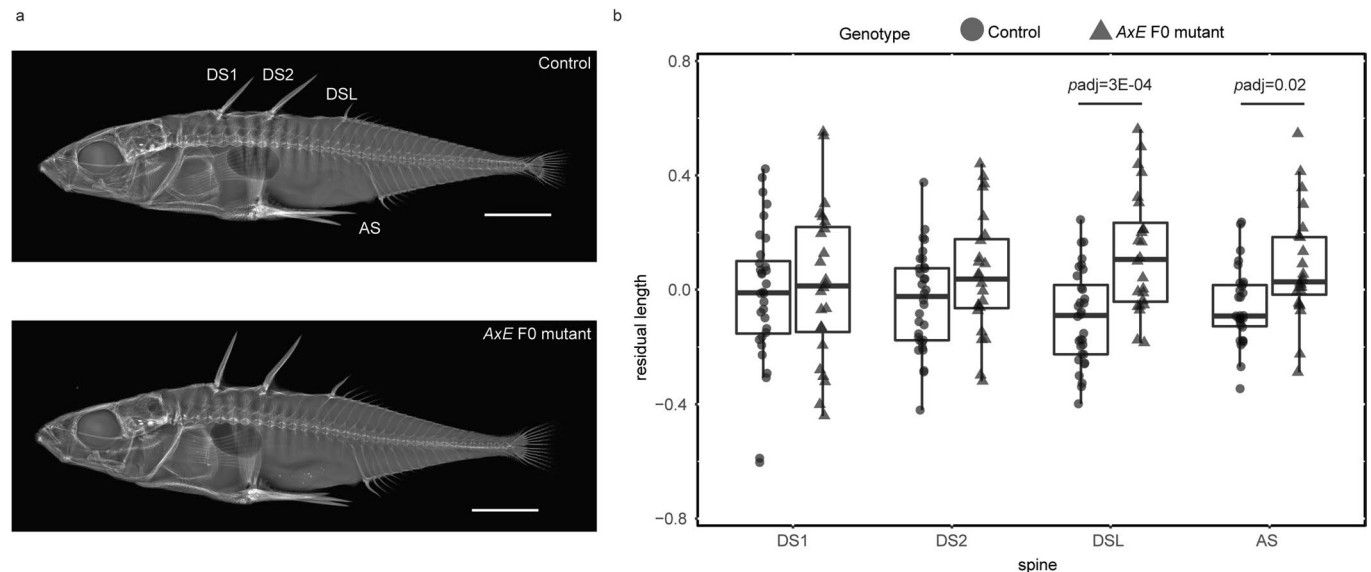

**Extended Data Fig. 9 | Targeting of *AxE* in an anadromous *Gasterosteus* population also causes length changes in DSL and AS. a**. Representative uninjected LITC sibling control fish (top) and injected *AxE* F0 mutant (bottom). Scale bar is 5 mm. **b**. Quantification of spine length difference. In the box and whisker plot: center line, median; box limits, interquartile range; whiskers, 1.5x interquartile range; each measurement is represented by a single point (circle for wild type and triangle for mutant). The residual after adjusting for standard length is on the y-axis and the spines ordered from anterior to posterior are on the x-axis. DS1 and DS2 do not show a significant difference in length between control and injected. DSL and AS were significantly longer in the injected compared to the control (two-tailed t-test Bonferroni-corrected at α = 0.05, DSL *p*adj = 3E-04, AS *p*adj = 0.02 n = 32 control and n = 24 injected).

|  | G. aculeatus | G. wheatlandi | Culaea | Pungitius | Apeltes | Spinachia |
|---|---|---|---|---|---|---|
| G. aculeatus | ■ |  |  |  |  |  |
| G. wheatlandi | 0.76 | ■ |  |  |  |  |
| Culaea | 0.88 | 0.74 | ■ |  |  |  |
| Pungitius | 0.59 | 0.66 | 0.72 | ■ |  |  |
| Apeltes | **1.24** | 0.93 | 0.97 | 0.76 | ■ |  |
| Spinachia | 0.78 | 0.71 | 0.72 | 0.53 | 0.80 | ■ |

**Extended Data Fig. 10 | dN/dS values for *HOXD11B* between pairs of stickleback species.** The tree on the left shows phylogenetic relationships of extant stickleback species (branch lengths not drawn to scale,[30,109]). The rate of non-synonymous to synonymous substitutions in *HOXD11B* is higher than 1 for *Gasterosteus* and *Apeltes* comparisons (yellow shading).

**nature** portfolio

# Reporting Summary

## Statistics

For all statistical analyses, confirm that the following items are present in the figure legend, table legend, main text, or Methods section.

| n/a | Confirmed | |
|---|---|---|
| ☐ | ☒ | The exact sample size (*n*) for each experimental group/condition, given as a discrete number and unit of measurement |
| ☐ | ☒ | A statement on whether measurements were taken from distinct samples or whether the same sample was measured repeatedly |
| ☐ | ☒ | The statistical test(s) used AND whether they are one- or two-sided<br>*Only common tests should be described solely by name; describe more complex techniques in the Methods section.* |
| ☐ | ☒ | A description of all covariates tested |
| ☐ | ☒ | A description of any assumptions or corrections, such as tests of normality and adjustment for multiple comparisons |
| ☐ | ☒ | A full description of the statistical parameters including central tendency (e.g. means) or other basic estimates (e.g. regression coefficient) AND variation (e.g. standard deviation) or associated estimates of uncertainty (e.g. confidence intervals) |
| ☐ | ☒ | For null hypothesis testing, the test statistic (e.g. *F*, *t*, *r*) with confidence intervals, effect sizes, degrees of freedom and *P* value noted<br>*Give P values as exact values whenever suitable.* |
| ☒ | ☐ | For Bayesian analysis, information on the choice of priors and Markov chain Monte Carlo settings |
| ☒ | ☐ | For hierarchical and complex designs, identification of the appropriate level for tests and full reporting of outcomes |
| ☐ | ☒ | Estimates of effect sizes (e.g. Cohen's *d*, Pearson's *r*), indicating how they were calculated |

*Our web collection on statistics for biologists contains articles on many of the points above.*

## Software and code

Policy information about availability of computer code

| Data collection | See below. |
|---|---|
| Data analysis | The custom code, programs (versions and associated parameters) used to analyze the data is available at figshare and details are provided in the materials and methods. |

For manuscripts utilizing custom algorithms or software that are central to the research but not yet described in published literature, software must be made available to editors and reviewers. We strongly encourage code deposition in a community repository (e.g. GitHub). See the Nature Portfolio guidelines for submitting code & software for further information.

## Data

Policy information about availability of data

All manuscripts must include a data availability statement. This statement should provide the following information, where applicable:
- Accession codes, unique identifiers, or web links for publicly available datasets
- A description of any restrictions on data availability
- For clinical datasets or third party data, please ensure that the statement adheres to our policy

The raw and processed RNA-sequencing data in this paper are available in the NCBI GEO database: GSE184888 (subseries GSE184885, GSE184886, GSE184887, GSE190498). The PacBio HiFi and 10X linked read data from Gasterosteus high-spine sequencing are available in the NCBI databases under BioProject number: PRJNA766710. The 10X linked read data from Apeltes quadracus four- and five-spine fish are available under BioProject number: PRJNA769115. The sequence surrounding AxE in Gasterosteus from the two parental QTL populations and the Apeltes AxE sequences tested in transgenic assays are available in GenBank (OK383406, OK383407, OK383404, OK383405). QTL mapping files, phenotype data files, association mapping genotype files, and code are available at Figshare. The pTia1l-hspGFP plasmid is available from Addgene. Other materials will be made available upon request.

# Field-specific reporting

Please select the one below that is the best fit for your research. If you are not sure, read the appropriate sections before making your selection.

☐ Life sciences　　☐ Behavioural & social sciences　　☒ Ecological, evolutionary & environmental sciences

For a reference copy of the document with all sections, see nature.com/documents/nr-reporting-summary-flat.pdf

# Ecological, evolutionary & environmental sciences study design

All studies must disclose on these points even when the disclosure is negative.

| | |
|---|---|
| Study description | We conducted QTL mapping in threespine sticklebacks (Gasterosteus aculeatus) and association mapping in four spine sticklebacks (Apeltes quadracus) for phenotypes relating to the axial patterning of sticklebacks including the dorsal spines.  We focused on the role of one Hox cluster and examined the gene expression changes that affect the patterning of these axial traits. |
| Research sample | We collected threespine sticklebacks (Gasterosteus aculeatus) and fourspine stickleback (Apeltes quadracus) from the populations described in Supplementary table 1.  All fish raised in the animal facility at Stanford were treated in accordance with the recommendations in the Guide for the Care and Use of Laboratory Animals of the National Institutes of Health, using protocols approved by the Institutional Animal Care and Use Committee of Stanford University (IACUC protocol #13834), in animal facilities accredited by the Association for Assessment and Accreditation of Laboratory Animal Care International (AAALAC). |
| Sampling strategy | We collected fish that represented the whole range of phenotypes present in the population.  When choosing fish to use for the QTL cross, fish with the most extreme phenotypes that were sexually mature were chosen.  For RNA-sequencing and expression experiments, fish were chosen based on their reproductive condition and their genotype at the locus of interest. |
| Data collection | Field sampling was conducted by JIW, TRH, ALH, TER, MAB, and ACD with minnow traps at the locations detailed in supplemental table 1.  The QTL mapping genotype data was collected by TRH.  All RNA-sequencing data was collected by JIW.  Whole genome DNA sequencing data was collected by JIW and EHA. |
| Timing and spatial scale | Samples for Gasterosteus QTL mapping were collected in 2009; Samples for Apeltes association mapping were collected in 2018 and 2019. Sequence and phenotypic analyses was performed between 2011 and 2021. |
| Data exclusions | Individuals from the Gasterosteus QTL mapping were excluded from the downstream analysis based on the quality of the genotyping calls from the SNP Chip data as detailed in the materials and methods under "QTL mapping". |
| Reproducibility |  Data analysis is fully reproducible and all raw genomic data, phenotypic measurements, processed files, parameters, and code necessary are provided as supplementary material. |
| Randomization | For morphological association studies, wild caught fish were assigned to high-spined or low-spined groups based on counting spines in skeletal X-rays.   Animal numbering within groups was random, and all animals were include unless they showed evidence of broken spines that would preclude accurate measurements. |
| Blinding | Phenotype measurements for QTL mapping and association mapping were performed without knowing the genotype information for the given individual. |

Did the study involve field work?　☒ Yes　☐ No

## Field work, collection and transport

| | |
|---|---|
| Field conditions | Samples were collected in the late spring and early summer when the fish were in reproductive condition.  They were caught using minnow traps, seine nets, or dip nets depending on which was appropriate and approved under the relevant permit. |
| Location | The locations and species sampled are provided in supplemental table 1. |
| Access & import/export | Fieldwork has been conducted in compliance with appropriate national laws of each country (USA and Canada) and using permits obtained by the appropriate regional or state authorities. |
| Disturbance | No disturbance was caused by this study. |

# Reporting for specific materials, systems and methods

We require information from authors about some types of materials, experimental systems and methods used in many studies. Here, indicate whether each material, system or method listed is relevant to your study. If you are not sure if a list item applies to your research, read the appropriate section before selecting a response.

## Materials & experimental systems

| n/a | Involved in the study |
|---|---|
| ☒ | ☐ Antibodies |
| ☒ | ☐ Eukaryotic cell lines |
| ☒ | ☐ Palaeontology and archaeology |
| ☐ | ☒ Animals and other organisms |
| ☒ | ☐ Human research participants |
| ☒ | ☐ Clinical data |
| ☒ | ☐ Dual use research of concern |

## Methods

| n/a | Involved in the study |
|---|---|
| ☒ | ☐ ChIP-seq |
| ☒ | ☐ Flow cytometry |
| ☒ | ☐ MRI-based neuroimaging |

# Animals and other organisms

Policy information about studies involving animals; ARRIVE guidelines recommended for reporting animal research

| Laboratory animals | Wild sticklebacks were collected in the field, and their offspring were raised in the animal facility at Stanford University. |
|---|---|
| Wild animals | Wild sticklebacks were captured using minnow traps, dip nets, or small minnow seines. The populations used for this study and their GPS coordinates are listed in Table S1. |
| Field-collected samples | Fish husbandry was done using standard methods for sticklebacks. The fish are housed in the animal facility at Stanford under controlled temperature and lighting conditions. The fish are feed twice daily with a mix of blood worms, mysis shrimp, cyclops, and artemia, and the water quality and health checks are performed daily. |
| Ethics oversight | All sticklebacks were treated in accordance with the recommendations in the Guide for the Care and Use of Laboratory Animals of the National Institutes of Health, using protocols approved by the Institutional Animal Care and Use Committee of Stanford University (IACUC protocol #13834), in animal facilities accredited by the Association for Assessment and Accreditation of Laboratory Animal Care International (AAALAC). |

Note that full information on the approval of the study protocol must also be provided in the manuscript.

