## [Peer Review File · Nature Ecology & Evolution]

Peer Review Information

Journal: Nature Ecology & Evolution

Manuscript Title: Evolution of stickleback spines through independent cis-regulatory changes at HOXDB

Corresponding author name(s): David M. Kingsley

Editorial Notes:

Reviewer Comments & Decisions:

Decision Letter, initial version:
--

31st March 2022

Dear Dr Kingsley,

Your manuscript entitled "Evolution of stickleback spines through independent *cis*-regulatory changes at *HOXDB*" has now been seen by 3 reviewers, whose comments are attached. The reviewers have raised a number of concerns which will need to be addressed before we can offer publication in Nature Ecology & Evolution. We will therefore need to see your responses to the criticisms raised and to some editorial concerns, along with a revised manuscript, before we can reach a final decision regarding publication.

* If you have not done so already please begin to revise your manuscript so that it conforms to our Article format instructions at <http://www.nature.com/natecolevol/info/final-submission>. Refer also to any guidelines provided in this letter.

[REDACTED]

Nature Ecology & Evolution is committed to improving transparency in authorship. As part of our efforts in this direction, we are now requesting that all authors identified as 'corresponding author' on published papers create and link their Open Researcher and Contributor Identifier (ORCID) with their account on the Manuscript Tracking System (MTS), prior to acceptance. ORCID helps the scientific community achieve unambiguous attribution of all scholarly contributions. You can create and link your ORCID from the home page of the MTS by clicking on 'Modify my Springer Nature account'. For more information please visit www.springernature.com/orcid.

[REDACTED]

Reviewers' comments:

Reviewer #1 (Remarks to the Author):

Comments to the authors

It has been discussed for a long time how the Hox genes, one of the most fundamental developmental genes, contribute to the diversity of morphology. In this study, the authors investigated the genetic mechanisms underlying the variations in length and number of dorsal spines of stickleback species and revealed that the same HOXD genes but different cis-regulatory changes provide the evolution of adaptive skeletal variations in two distinct groups of sticklebacks.

The experiments are well designed, the manuscript is well written, and the data are clearly and logically presented with a few minor exceptions (see Specific comments). Their findings are fruitful to understand the new aspect of the evolutionary importance of Hox genes to contribute to diversification. However, I have several comments before the publication.

[Major comments]

Although the authors explain the adaptive significance of dorsal spine number and length in the discussion part, they did not investigate whether the cis-regulatory region of HOXD11B is under the positive selection in *Gasterosteus* and *Apeltes*. To conclude whether the cis-regulatory changes of HOXD gene contribute to adaptive evolution, the authors need to detect the selection in the genomic regions or conduct the predation experiments using CRISPR/Cas9 individuals.

[Specific comments]

Line 120-123: Which dorsal spine has been lost in the Boulton Lake stickleback? The authors do not show any actual photos of variations of sticklebacks. It would be better to show them in the supplementary figure.

Line 125-126: It also would be better to show the photos of stickleback with two, three, and four spines.

Line 139-143: What proportion of phenotypic variance for three- vs four-spine categorical traits, and DS2 length were explained by the QTL on chromosome 6?

Line 166-168: It is difficult to understand where is the neural tube, and anterior-most somite in Supplementary Fig. 2. Please point to it in the figure.

Line 221-222: How about the function of other HOXD genes in the cluster? If the authors only focused on HOXD11B and conducted the functional analysis, the title of this section should be "HOXD11B is in the candidate interval and expressed in *Gasterosteus* spines".

Line 232-233: Why the F1 hybrids show the variation of the number of spines? Did the spine length also show such variations?

Line 241-242: Where is the 5 different informative SNP located in the region? It would be better to show it as a supplementary figure.

Line 291-293: It also would be worth showing the variations of *Apeltes* with two to six spines in the supplementary figure.

Line 364-365: Did the authors conduct the statistical analysis for the phenotype?

3Line 418-422: The authors should conduct the statistical analysis.

[Minor comments]

- Most of the "p =" for p-value should be in Italic.

Reviewer #2 (Remarks to the Author):

Using a mixture of QTL mapping, association mapping, and functional genetics, the authors identify that variation in spine length and spine number across populations and species of stickleback fish is partly due to the convergent evolution of regulatory regions within the HOXDB locus. These are important results that add to a growing body of literature that Hox gene variation underlies body plan evolution. In addition, this work provides another compelling example that convergent evolution can be due to repeated mutation at the same genetic locus. These results will surely be of broad interest to the evolutionary and developmental biology communities. The functional genetic work in this paper is very convincing. My main comments are for clarification for some of the interpretations.

Although one strength of this paper is providing evidence from multiple populations and species, this also quickly becomes a challenge when trying to interpret all the different patterns. It took many readthroughs to reconcile the mechanism that results in the gain of spines in *Gasterosteus* and *Apeltes*. For instance, in the main figure, the high-spine phenotype in *Gasterosteus* is achieved by expanding the HOXDB expression domain, decreasing DS2 length and creating a new spine (DS3) through a new posterior-like, DSL identity. However, the high-spine phenotype in *Apeltes* is achieved by reducing the expression domain of HOXDB, increasing DS3 length and creating a new DS4 spine through anteriorization. I think a couple things could help clarify these opposite expression domains in the text. For one, in the discussion it would be helpful to propose a mechanism how a gain of a TF binding site in *Apeltes* leads to reduced HOXDB expression, whereas deletion of an enhancer in *Gasterosteus* leads to expanded HOXDB expression. Also, in the first main figure and early in the text, the authors should clearly highlight that gain of spines and alteration in spine length can be achieved by both anteriorization and posteriorization. Longer spines are a pattern of anteriorization and shorter spines are a product of posteriorization. Both of these scenarios could be labeled directly in the figure. It would also be helpful to label where the "blank pterygiophores" are.

Another underexplained aspect of this paper is with the initial QTL mapping. The authors cross two different populations of threespine sticklebacks: Boulton Lake (two dorsal spines) with Bodega Bay (three dorsal spines). The main QTL that the authors present is on chromosome 6. The allele that causes an increase in spine number and a decrease in DS2 length (posteriorization) is actually from the Boulton Lake population, which predominantly has fewer spines. Does this mean the initial female that was used for the cross from Boulton Lake was heterozygous for a low and high spine allele? Furthermore, the phenotype that was associated with chromosome 6, was binary: three versus four spines. As reported by the authors, the four spine configuration is never found in the Boulton Lake population and this represents a transgressive phenotype. This combined with the fact that deleting

4the AxE sequence alone does not produce a high spine phenotype, suggests this is a polygenic trait. Do the authors think that other enhancer regions underlying the chromosome 6 QTL are responsible? And/or are other regions in the genome? It would be helpful to know how much of the phenotypic variation is explained by the chromosome 6 QTL. This data should be reported for all the phenotypes mapped in the QTL experiment.

The authors also propose that the deletion in AxE in Boulton Lake is likely under selection within this population because it produces shorter dorsal spines (lines 510-512). However, at least in the QTL mapping population, this allele is also associated with increase spine number from three to four. Yet, no fish in the population have four spines and actually the two spine phenotype is at a frequency of 80%. This suggests the low spine allele is under selection? How do the authors resolve these two separate phenotypes (spine length and spine number) in this model?

Other Comments:

Lines 34-36: “. . . but do so by diverse mechanisms, including SNPs, deletions, and transposable elements.” The authors find transposable elements around the AxE deletion and also note transposable elements are also around this location in other populations, even if there is not a large deletion. To my understanding, there is no functional evidence that the TE’s contribute to this phenotype. They are just associated with the region. In the populations where the authors noted TE’s and no deletions are there changes in spine length and spine number? I think it is speculative to state TE’s are involved.

Lines 134-135: The authors genotyped only 340 fish from the total cross of 590. How were these 340 chosen? There is no justification in the manuscript for this subsample.

Reviewer #3 (Remarks to the Author):

In their work Wucherpfenning et al. identified the HOXDB locus to be linked to spine length and number variation. The amount and quality of data with analyses in different species, quantitative trait loci mapping, allele-specific expression, CRISPR and transgenics is quite remarkable and through elegant combination of these techniques very convincing. The manuscript, despite being quite complex, is very well written. What makes it complex is maybe also the combination of the two traits, number and length, but I think it is nice and good to have them discussed together. Same is true with the two genera. I really enjoyed the historical perspective in the introduction that very much demonstrates why this work is important and deserves publication. I believe this could become a text book example for effects of hox gene regulatory evolution in vertebrates much of which only has been speculated about based on model organisms knockouts. Congratulations.

I only have a few points mainly one the QTL part that I would believe to further improve the manuscript.

Specific comments:

51. It would be nice to see the trait distribution for the dorsal spine length (at least spine 2) of the parental populations, the F2s (and the F1s if possible). This could be nicely fit into Figure 1.
2. Furthermore a plot that shows the distribution of the spine numbers (2,3,4 spines using stacked bar plots) but also the spine length phenotype by genotype within the F2 would be helpful.

Additional minor comments.

L125: Maybe be specific and add the percentage or number of F2s with three spines.

L140: Should it not be binary trait if you only have two or three? Did you try to simply map the trait as quantitative trait with the values 2,3 and 4?

L143: I am confused by the fact that the increase in spine number is linked to the allele of the population with has less spines. Can the authors comment on that. I am wondering if the regular QTL analysis is appropriate if one deals with a transgressive phenotype that behaves opposite to the genotype-phenotype relationship shown in the grandparents — if I understood correctly. Maybe the authors can also comment on this.

L170: Can you give a bit more reasoning for this strategy. I am familiar with enhancer-GFP constructs and knock-ins that would replace the ORF of the target gene with GFP. Is the reasoning here that the activity of GFP with a ubiquitous promoter would be affected by the epigenetic state of the hox cluster and therefore be a proxy for *hoxd11b* expression? Or is it activation by cis-regulatory elements at this locus? I understand why you do this, but not why you chose this particular approach.

Figure 2: Maybe indicate somewhere that this is done on "low-spine sticklebacks" o make it easier for the reader.

Figure 2E: It is fine on the pdf, but additional photos with higher magnification might be good to have in this figure.

L221: Would it make sense to emphasise here that spine number is not affected?r

L222: Can DSL length and pterygiophore number really be considered as "patterning of the dorsal spines"? Patterning I would always consider as the development of different identities within a organ/tissue/element.

L416: It is too bad that no differences in expression patterns and strength, but I agree with the authors that this is not unexpected due to the use of Tol2 integration and mosaicism and does not exclude that differences in strength and pattern exist.

*****END*****

Author Rebuttal to Initial comments

We have revised the manuscript to address the useful comments and suggestions of all three reviewers. We list the detailed revisions below, showing the original reviewer comments in black text, and our responses and changes in red. Note that line numbers described in red text below refer to line numbers in the PDF version of the revised manuscript. We are also providing a Word file with "track changes" on,

6but line numbers in that document sometimes vary when opened with different versions of Microsoft Word.

Reviewer 1:

It has been discussed for a long time how the Hox genes, one of the most fundamental developmental genes, contribute to the diversity of morphology. In this study, the authors investigated the genetic mechanisms underlying the variations in length and number of dorsal spines of stickleback species and revealed that the same HOXD genes but different cis-regulatory changes provide the evolution of adaptive skeletal variations in two distinct groups of sticklebacks.

The experiments are well designed, the manuscript is well written, and the data are clearly and logically presented with a few minor exceptions (see Specific comments). Their findings are fruitful to understand the new aspect of the evolutionary importance of Hox genes to contribute to diversification. However, I have several comments before the publication.

Thank you for your positive evaluation of the design, presentation, and overall significance of the studies.

[Major comments]

Although the authors explain the adaptive significance of dorsal spine number and length in the discussion part, they did not investigate whether the cis-regulatory region of HOXD11B is under the positive selection in *Gasterosteus* and *Apeltes*. To conclude whether the cis-regulatory changes of HOXD gene contribute to adaptive evolution, the authors need to detect the selection in the genomic regions or conduct the predation experiments using CRISPR/Cas9 individuals.

Although extensive previous studies have established that dorsal spine phenotypes in sticklebacks are adaptive, we agree that we have not shown directly that the specific *Hox* alleles identified in this study have also been under positive selection. We have modified the text at two points to further clarify this point, and to discuss the prospects for future studies:

1) In the first paragraph of the discussion (lines 532-535), we reworded a sentence in which it was previously not clear whether “adaptive” referred to *Hox* alleles specifically, or spine phenotypes in general. The revised sentence reads: “Our studies show that independent regulatory changes have

7occurred in the *HOXDB* locus of *Gasterosteus* and *Apeltes*, providing a compelling example of *cis*-acting variation in *Hox* genes linked to the evolution of novel axial skeletal patterns in wild vertebrate species.”

2) We also added a new paragraph in our discussion of “Adaptive significance of dorsal spine number and length”. The new paragraph (lines 585-597) states:

“Although spine phenotypes are clearly adaptive in sticklebacks, we note that we have not yet established whether the specific *HOXDB* alleles identified in the Boulton and Nova Scotia populations have been subject to positive selection. Multiple dorsal spine phenotypes have been present in the Boulton, Louisbourg and Tidnish populations for decades, likely due to long-term balancing selection in the face of fluctuating predation regimes^{49,60,67}. Long-term balancing selection is more difficult to detect at the molecular level than selective sweeps of a particular favored allele⁶⁹. However, recent experiments have successfully monitored short-term changes in the frequency of particular alleles of interest in sticklebacks following changes in environmental conditions^{54,70} or experimental exposure to predators⁵⁹. In the future, it will be interesting to extend such studies to differential survival of particular *Hox* alleles and dorsal spine phenotypes, using either the alternative alleles we have identified here from natural *Gasterosteus* and *Apeltes* populations, or transgenic and CRISPR-edited sticklebacks engineered to carry particular sequence variants at the *HOXDB* locus.”

We also added two associated references to the bibliography:

69. Vitti, J.J., Grossman, S.R., and Sabeti, P.C. (2013). Detecting natural selection in genomic data. *Annual Review of Genetics* 47, 97–120. <https://doi.org/10.1146/annurev-genet-111212-133526>.

70. Barrett, R.D.H., Rogers, S.M., and Schluter, D. (2008). Natural selection on a major armor gene in threespine stickleback. *Science* 322, 255–257. <https://doi.org/10.1126/science.1159978>.

[Specific comments]

Line 120-123: Which dorsal spine has been lost in the Boulton Lake stickleback? The authors do not show any actual photos of variations of sticklebacks. It would be better to show them in the supplementary figure.

We have further clarified these points in Supplementary Figure 1 as requested. The revised figure now includes x-ray images of the parents of the cross, and further description of phenotypes in the legend. Note that the remaining dorsal spine in Boulton Lake is located at an intermediate position relative to

the typical positions of DS1 and DS2, in a three-spine fish (Reimchen, 1980; doi: 10.1139/z80-173), and it is therefore referred to as DS in the figure.

Line 125-126: It also would be better to show the photos of stickleback with two, three, and four spines.

As requested, we have added examples of F2 fish with two, three and four spines to Supplementary Figure 1.

Line 139-143: What proportion of phenotypic variance for three- vs four-spine categorical traits, and DS2 length were explained by the QTL on chromosome 6?

We have added the percent variance explained to the text (lines 144-145).

Line 166-168: It is difficult to understand where is the neural tube, and anterior-most somite in Supplementary Fig. 2. Please point to it in the figure.

We have added labels to clarify the location of the hindbrain, anterior somites, and neural tube (now Supplementary Figure 3).

Line 221-222: How about the function of other HOXD genes in the cluster? If the authors only focused on HOXD11B and conducted the functional analysis, the title of this section should be “HOXD11B is in the candidate interval and expressed in *Gasterosteus* spines”.

We have changed the title of this section as suggested by the reviewer (now line 179).

Line 232-233: Why the F1 hybrids show the variation of the number of spines?

The parents of the F1 experimental crosses are not from inbred populations. In particular, the high-spine male parent was an F5 generation offspring from the BOUL/BDGB QTL cross. We genotyped this fish to ensure that all of its own F1 offspring would carry the Boulton allele at the *HOXD* locus. However, the offspring still carry a mixture of Boulton and Bodega Bay alleles at other loci in the genome. Given that the *HOXD* locus only accounts for a proportion of total variance explained, other loci that are also segregating in the fish can also affect dorsal spine number. Fortunately, allele-specific expression

9experiments compare the relative level of expression of two alleles within individual fish, rather than between fish, so this cross design is still highly effective for detecting possible *cis*-regulatory effects at the *Hox* loci of interest.

Line 232-233: Did the spine length also show such variations?

Because fish were dissected at juvenile stages (11-13 mm in body length), the spines were too small (less than one millimeter in length) to assess quantitative variation in length.

Line 241-242: Where is the 5 different informative SNP located in the region? It would be better to show it as a supplementary figure.

As suggested, we have added a new Supplementary Figure 6 to show where the SNPs are located in the *HOXDB* genes and which SNPs were informative for allele-specific expression measurements in the experimental crosses.

Line 291-293: It also would be worth showing the variations of *Apeltes* with two to six spines in the supplementary figure.

As requested, we have added Supplementary Figure 8 to show the spine number variation in Tidnish *Apeltes*, where spine number ranges from two to six spines.

Line 364-365: Did the authors conduct the statistical analysis for the phenotype?

We have added a statistical analysis of differences seen between the genotypic classes (now lines 367-370).

Line 418-422: The authors should conduct the statistical analysis.

We have added a statistical analysis (now lines 427-428).

[Minor comments]

- Most of the “p =” for p-value should be in Italic.

We have changed the *p*-values to italics as requested.

Reviewer #2 (Remarks to the Author):

Using a mixture of QTL mapping, association mapping, and functional genetics, the authors identify that variation in spine length and spine number across populations and species of stickleback fish is partly due to the convergent evolution of regulatory regions within the HOXDB locus. These are important results that add to a growing body of literature that Hox gene variation underlies body plan evolution. In addition, this work provides another compelling example that convergent evolution can be due to repeated mutation at the same genetic locus. These results will surely be of broad interest to the evolutionary and developmental biology communities. The functional genetic work in this paper is very convincing. My main comments are for clarification for some of the interpretations.

Thank you for the positive comments about the importance and wide interest of the studies.

Although one strength of this paper is providing evidence from multiple populations and species, this also quickly becomes a challenge when trying to interpret all the different patterns. It took many readthroughs to reconcile the mechanism that results in the gain of spines in *Gasterosteus* and *Apeltes*. For instance, in the main figure, the high-spine phenotype in *Gasterosteus* is achieved by expanding the HOXDB expression domain, decreasing DS2 length and creating a new spine (DS3) through a new posterior-like, DSL identity. However, the high-spine phenotype in *Apeltes* is achieved by reducing the expression domain of HOXDB, increasing DS3 length and creating a new DS4 spine through anteriorization. I think a couple things could help clarify these opposite expression domains in the text. For one, in the discussion it would be helpful to propose a mechanism how a gain of a TF binding site in *Apeltes* leads to reduced HOXDB expression, whereas deletion of an enhancer in *Gasterosteus*

leads to expanded HOXDB expression. Also, in the first main figure and early in the text, the authors should clearly highlight that gain of spines and alteration in spine length can be achieved by both anteriorization and posteriorization. Longer spines are a pattern of anteriorization and shorter spines

11are a product of posteriorization. Both of these scenarios could be labeled directly in the figure. It would also be helpful to label where the “blank pterygiophores” are.

We agree it is challenging to keep track of multiple results and anatomical patterns across different alleles and populations, though the overall combination of results is also one of the main strengths of the study. To further aid the reader, we have now added a new summary and model figure (Figure 6) which efficiently summarizes how we think genotypes, expression patterns, and phenotypes are linked. As suggested by the reviewer, we label anteriorization and posteriorization directly in this summary and also indicate the position of pterygiophores that can be either blank or spine-bearing in different fish.

To accommodate this useful additional figure, while still adhering to the journal limit of six figures in the main manuscript, we combined data from parts of previous Figures 1 and 2 to make a revised Figure 1, moved embryonic GFP expression data from previous Figure 2 to a revised Supplementary Figure 3 (where it can now be compared directly to embryonic *in situ* patterns). We have also renumbered all figures appropriately throughout the manuscript.

Another underexplained aspect of this paper is with the initial QTL mapping. The authors cross two different populations of threespine sticklebacks: Boulton Lake (two dorsal spines) with Bodega Bay (three dorsal spines). The main QTL that the authors present is on chromosome 6. The allele that causes an increase in spine number and a decrease in DS2 length (posteriorization) is actually from the Boulton Lake population, which predominantly has fewer spines. Does this mean the initial female that was used for the cross from Boulton Lake was heterozygous for a low and high spine allele? Furthermore, the phenotype that was associated with chromosome 6, was binary: three versus four spines. As reported by the authors, the four spine configuration is never found in the Boulton Lake population and this represents a transgressive phenotype. This combined with the fact that deleting the AxE sequence alone does not produce a high spine phenotype, suggests this is a polygenic trait. Do the

authors think that other enhancer regions underlying the chromosome 6 QTL are responsible? And/or are other regions in the genome? It would be helpful to know how much of the phenotypic variation is explained by the chromosome 6 QTL. This data should be reported for all the phenotypes mapped in the QTL experiment.

As requested, we have added the percent variance explained for both the 4-spine trait and the length of DS2 (lines 144-145). Given that the percent variance explained is only 6 and 8% at the distal end of chr6 in our QTL cross, there are likely many other loci that also contribute. Some of the other spine length

loci on chr4 have previously been studied, and we provide additional information and relevant references in the accompanying figure legend (lines 159-160). To our knowledge this is the first paper that has studied increased spine numbers in either *Gasterosteus* or *Apeltes*, and the chr6 locus is the only region that passes genome-wide significance in our cross, even though it only explains a portion of the overall variance in the trait.

We also describe in the manuscript that deletion of *AxE* on a low-spine background is not sufficient to recapitulate the four-spine trait (lines 449-458). Our CRISPR experiments deleting *AxE* (renumbered to Supplementary Figure 10) show that the F0 mutants do not have a spine number phenotype. Because the Boulton allele consists of both a deletion of *AxE* and an insertion of additional transposable elements, we think that the insertion of the endogenous retrovirus (ERV) may also be required to reproduce the four-spine trait. While we are not able to test this by recreating the entire complex deletion/insertion allele using CRISPR, we have now added new data showing that the long terminal repeat (LTR) of the ERV shows enhancer activity in developing stickleback larvae (lines 460-476). This provides experimental evidence that the added sequences contain new regulatory information (see additional information under “other comments” below).

The authors also propose that the deletion in *AxE* in Boulton Lake is likely under selection within this population because it produces shorter dorsal spines (lines 510-512). However, at least in the QTL mapping population, this allele is also associated with increase spine number from three to four. Yet, no fish in the population have four spines and actually the two spine phenotype is at a frequency of 80%. This suggests the low spine allele is under selection? How do the authors resolve these two separate phenotypes (spine length and spine number) in this model?

We believe that the four-spine phenotype in the QTL cross only appears in the context of other alleles inherited from the marine fish in the Boulton cross. The phenomenon of F2 progeny exceeding the phenotype of either F0 parent is common and has also been observed in other crosses. We have included a review of “transgressive variation” for interested readers in summarizing these results (lines 553-556, plus reference 65).

We also now further comment on the non-intuitive effects of the Boulton allele in lines 557-563 of the discussion:

13“We hypothesize that a *HOXDB* allele likely evolved in this population for its contributions to reduced DS2 length via a posteriorizing mechanism in the wild lake population (Figure 6). In the Boulton genetic background, this posteriorizing effect may be sufficient to change DS length without inducing formation of a new spine on a blank pterygiophore, while in the mixed genetic background of the QTL cross, the posteriorizing tendency of the Boulton allele appears to lead to both spine length and spine number changes.”

Finally, as noted above in the response to reviewer 1, we have also added additional information on selection on spine phenotypes to the Discussion (lines 585-597). This additional information includes references to previous fluctuating selection in Boulton Lake and mentions the types of experiments that could be carried out in the future to try to test whether *Hox* alleles confer an advantage under particular conditions.

Other Comments:

Lines 34-36: “. . . but do so by diverse mechanisms, including SNPs, deletions, and transposable elements.” The authors find transposable elements around the AxE deletion and also note transposable elements are also around this location in other populations, even if there is not a large deletion. To my understanding, there is no functional evidence that the TE’s contribute to this phenotype. They are just associated with the region. In the populations where the authors noted TE’s and no deletions are there changes in spine length and spine number? I think it is speculative to state TE’s are involved.

To investigate whether sequences from the transposable element have added regulatory information to the Boulton allele, we have now carried out additional enhancer assays in transgenic fish. We cloned either the LINE element, or the long terminal repeat (LTR) from the endogenous retrovirus (ERV) found in the Boulton allele into a *To12* expression vector with a basal promoter and GFP reporter gene. Injecting these constructs into sticklebacks showed that while the empty vector and the LINE element did not show enhancer activity, the LTR sequence drove consistent expression patterns in the posterior muscle, dorsal spines, dorsal fin, anal fin, heart, and gills. These results confirm that the new sequences in Boulton have enhancer activity. Because we are already at the maximum number of allowed Figures in both the Main text and Supplement, we describe these results by text in a new last paragraph of the results section, along with an appropriate statistical test of significance (lines 460-476):

“To test whether any of the additional transposable element sequences found in the Boulton high-spine allele might contribute new enhancer activities, we tested whether the ~1kb LINE, or the ~1kb long terminal repeat (LTR) of the endogenous retrovirus (ERV) could drive GFP reporter gene expression in transgenic enhancer assays compared to an empty vector control. While the empty vector and the LINE did not drive expression at early fin fold stage (Swarup stage 29), the LTR did drive expression in the dorsal fin fold, anal fin, posterior muscle, heart and gills (empty vector: n=17 transgenics with bilateral green eyes; n=2/17 whole body, n=1/17 heart, n=0/17 posterior muscle, n=0/17 dorsal fin fold, n=0/17 anal fin fold, n=0/17 gill, n=0/17 caudal fin; LINE: n=13 transgenics with bilateral green eyes; n=0/13 whole body, n=0/13 heart, n=0/13 posterior muscle, n=0/13 dorsal fin fold, n=0/13 anal fin fold, n=0/13 gill, n=0/13 caudal fin; LTR n=28 transgenics with bilateral green eyes; n=3/28 whole body, n=10/28 heart, n=16/28 posterior muscle, n=13/28 dorsal fin fold, n=12/28 anal fin fold, n=4/28 gill, n=16/28 caudal fin; Fisher’s exact test empty vector v. LTR, whole body $p_{adj}=1$, heart $p_{adj}=0.2$, posterior muscle $p_{adj}=4E-4$, dorsal fin fold $p_{adj}=4E-3$, anal fin fold $p_{adj}=7E-3$, gill $p_{adj}=1$, and caudal fin $p_{adj}=4E-4$). The retrovirus insertion that is unique to the Boulton allele thus includes new *cis*-regulatory enhancer sequences. Additional regulatory sequences may also be present in the rest of the insertion or surrounding region.”

Lines 134-135: The authors genotyped only 340 fish from the total cross of 590. How were these 340 chosen? There is no justification in the manuscript for this subsample.

Not all of the fish had high-quality data from the SNP chip and we filtered out individuals that had fewer than 600 genotype calls. In the revised manuscript, we clarify this difference in the text (lines 137-138), revised Figure 1 to eliminate potential confusion over numbers, and we provide full details in the methods for interested readers (lines 887-915)

Reviewer #3 (Remarks to the Author):

In their work Wucherpfenning et al. identified the HOXDB locus to be linked to spine length and number variation. The amount and quality of data with analyses in different species, quantitative trait loci mapping, allele-specific expression, CRISPR and transgenics is quite remarkable and through elegant combination of these techniques very convincing. The manuscript, despite being quite complex, is very well written. What makes it complex is maybe also the combination of the two traits, number and

15length, but I think it is nice and good to have them discussed together. Same is true with the two genera. I really enjoyed the historical perspective in the introduction that very much demonstrates why this work is important and deserves publication. I believe this could become a text book example for effects of hox gene regulatory evolution in vertebrates much of which only has been speculated about based on model organisms knockouts. Congratulations.

Thank you very much for this strong endorsement of the quality and significance of the studies.

I only have a few points mainly one the QTL part that I would believe to further improve the manuscript.

Specific comments:

1. It would be nice to see the trait distribution for the dorsal spine length (at least spine 2) of the parental populations, the F2s (and the F1s if possible). This could be nicely fit into Figure 1.

We have added a plot to show the distribution of DS2.res length and have included the parental fish from each population as colored points (see new Supplementary Fig. 1I).

2. Furthermore a plot that shows the distribution of the spine numbers (2,3,4 spines using stacked bar plots) but also the spine length phenotype by genotype within the F2 would be helpful.

We have added x-rays of the parental fish and now also show F2 progeny with 2, 3, or 4 spines in Supplementary Figure 1. We have also added a violin plot that shows the DS2 residual spine length phenotype broken out by different genotypes as requested (new Supplementary Fig. 1J).

Additional minor comments.

L125: Maybe be specific and add the percentage or number of F2s with three spines.

We have revised this sentence to read:

“Five hundred sixty-three F2 individuals had three dorsal spines, but six had two dorsal spines, and twenty-one had four dorsal spines (Supplementary Fig. 1).”

L140: Should it not be binary trait if you only have two or three? Did you try to simply map the trait as quantitative trait with the values 2,3 and 4?

We did try that (see below). We still saw a peak on distal chromosome 6, but neither that peak nor any of the other peaks detected passed a genome-wide significance threshold. This is likely because treating spine number as a quantitative trait required us to use a non-parametric testing model (because the distribution of 2,3, and 4 spines is not normally distributed) and the number of two-spine fish was very low.

L143: I am confused by the fact that the increase in spine number is linked to the allele of the population with has less spines. Can the authors comment on that. I am wondering if the regular QTL analysis is appropriate if one deals with a transgressive phenotype that behaves opposite to the genotype-phenotype relationship shown in the grandparents — if I understood correctly. Maybe the authors can also comment on this.

The reviewer is correct that the allele from the two-spine fish is linked to the four-spine trait. While this may seem non-intuitive, transgressive phenotypes are commonly seen in QTL crosses (Rieseberg et al., 1999, doi:10.1038/sj.hdy.6886170). The QTL analysis does not rely on the phenotype of the parent and only requires that phenotypic variation in the F2 progeny can be linked to genotypes in the F2 progeny. Therefore, standard QTL analysis can still be used, both here and in other studies (Rieseberg et al., 1999).

As discussed above in response to reviewer 2, we have also expanded lines 559-563 to provide additional comments on how a posteriorizing Boulton allele may produce different phenotypic effects on different backgrounds.

L170: Can you give a bit more reasoning for this strategy. I am familiar with enhancer-GFP constructs and knock-ins that would replace the ORF of the target gene with GFP. Is the reasoning here that the activity of GFP with a ubiquitous promoter would be affected by the epigenetic state of the hox cluster and therefore be a proxy for *hoxd11b* expression? Or is it activation by cis-regulatory elements at this locus? I understand why you do this, but not why you chose this particular approach.

We wanted to investigate the expression pattern of the gene without potentially disrupting the endogenous gene function. The approach of integrating a basal promoter plus reporter gene, which can then respond to nearby enhancers which also normally act on the endogenous locus, is similar to the “enhancer trap” approach that has been commonly used in *Drosophila* to study the expression patterns of many genes.

(O’Kane and Gehring, 1987, doi: 10.1073/pnas.84.24.9123; Singh, 1992, Current Science; https://www.jstor.org/stable/24096469?casa_token=HcrOqDTmGKMAAAA%3AA32m0wSWM5JMQ6s_wOqbhpDTLzWLAHjbBUiX6-i7InX0cupOGI33aO4WP_cvcNzzA2YFSnEJeZmPteonAzTNav1Hz-2xuvjfA_AgcpNeUwfINvH9agaA&seq=2).

Because we wanted to target a specific locus, we used CRISPR-Cas9 (as opposed to random integrations via transposable element) to target the region upstream of *HOXD11B*. This approach has previously been successfully used to generate reporter lines in zebrafish (Kimura et al., 2014, doi: 10.1038/srep06545). This allowed us to look at the expression pattern throughout development and allowed us to identify tissues and time points of expression that could then be validated by RNA-sequencing. Importantly, it did not require the insertion to be “perfect” or in-frame as this requirement would have dramatically reduced the rate of successful integrants, which is already much lower than generating random mutations via CRISPR-Cas9.

Figure 2: Maybe indicate somewhere that this is done on "low-spine sticklebacks" to make it easier for the reader.

We have added low-spine to the legend (lines 162-163). Due to figure number constraints, this is now found in revised Figure 1.

Figure 2E: It is fine on the pdf, but additional photos with higher magnification might be good to have in this figure.

We have added an inset with a higher magnification. Due to figure number constraints, these panels are now found in revised Figure 1.

L221: Would it make sense to emphasize here that spine number is not affected?

We have added a sentence that states "There was no effect on total spine number in either population." (lines 223-224).

L222: Can DSL length and pterygiophore number really be considered as "patterning of the dorsal spines"? Patterning I would always consider as the development of different identities within a organ/tissue/element.

We have removed the word patterning.

L416: It is too bad that no differences in expression patterns and strength, but I agree with the authors that this is not unexpected due to the use of Tol2 integration and mosaicism and does not exclude that differences in strength and pattern exist.

We concur with this assessment.

Decision Letter, first revision:

1924th May 2022

Dear Dr. Kingsley,

Thank you for submitting your revised manuscript "Evolution of stickleback spines through independent *cis*-regulatory changes at *HOXDB*" (NATECOLEVOL-220215728A). It has now been seen again by the original reviewers and their comments are below. The reviewers find that the paper has improved in revision, and therefore we'll be happy in principle to publish it in Nature Ecology & Evolution, pending minor revisions to satisfy the reviewers' final requests and to comply with our editorial and formatting guidelines.

[REDACTED]

Reviewer #2 (Remarks to the Author):

The authors have addressed all of my comments in detail. I have no further suggestions for the manuscript. This is exciting work and would be of broad interest to the readers of Nature Ecology & Evolution.

Reviewer #3 (Remarks to the Author):

The authors have carefully revised the manuscript to address all my concerns. Congratulations on this excellent work.

Our ref: NATECOLEVOL-220215728A

2026th May 2022

Dear Dr. Kingsley,

Thank you for your patience as we've prepared the guidelines for final submission of your Nature Ecology & Evolution manuscript, "Evolution of stickleback spines through independent *cis*-regulatory changes at *HOXDB*" (NATECOLEVOL-220215728A). Please carefully follow the step-by-step instructions provided in the attached file, and add a response in each row of the table to indicate the changes that you have made. Please also check and comment on any additional marked-up edits we have proposed within the text. Ensuring that each point is addressed will help to ensure that your revised manuscript can be swiftly handed over to our production team.

****We would like to start working on your revised paper, with all of the requested files and forms, as soon as possible (preferably within two weeks). Please get in contact with us immediately if you anticipate it taking more than two weeks to submit these revised files.****

In recognition of the time and expertise our reviewers provide to Nature Ecology & Evolution's editorial process, we would like to formally acknowledge their contribution to the external peer review of your manuscript entitled "Evolution of stickleback spines through independent *cis*-regulatory changes at *HOXDB*". For those reviewers who give their assent, we will be publishing their names alongside the published article.

Nature Ecology & Evolution offers a Transparent Peer Review option for new original research manuscripts submitted after December 1st, 2019. As part of this initiative, we encourage our authors to support increased transparency into the peer review process by agreeing to have the reviewer comments, author rebuttal letters, and editorial decision letters published as a Supplementary item. When you submit your final files please clearly state in your cover letter whether or not you would like to participate in this initiative. Please note that failure to state your preference will result in delays in accepting your manuscript for publication.

Cover suggestions

As you prepare your final files we encourage you to consider whether you have any images or illustrations that may be appropriate for use on the cover of Nature Ecology & Evolution.

21Covers should be both aesthetically appealing and scientifically relevant, and should be supplied at the best quality available. Due to the prominence of these images, we do not generally select images featuring faces, children, text, graphs, schematic drawings, or collages on our covers.

Nature Ecology & Evolution has now transitioned to a unified Rights Collection system which will allow our Author Services team to quickly and easily collect the rights and permissions required to publish your work. Approximately 10 days after your paper is formally accepted, you will receive an email in providing you with a link to complete the grant of rights. If your paper is eligible for Open Access, our Author Services team will also be in touch regarding any additional information that may be required to arrange payment for your article.

Please note that *Nature Ecology & Evolution* is a Transformative Journal (TJ). Authors may publish their research with us through the traditional subscription access route or make their paper immediately open access through payment of an article-processing charge (APC). Authors will not be required to make a final decision about access to their article until it has been accepted. [Find out more about Transformative Journals](https://www.springernature.com/gp/open-research/transformative-journals)

Authors may need to take specific actions to achieve [compliance](https://www.springernature.com/gp/open-research/funding/policy-compliance-faqs) with funder and institutional open access mandates. If your research is supported by a funder that requires immediate open access (e.g. according to [Plan S principles](https://www.springernature.com/gp/open-research/plan-s-compliance)) then you should select the gold OA route, and we will direct you to the compliant route where possible. For authors selecting the subscription publication route, the journal's standard licensing terms will need to be accepted, including [self-archiving-and-license-to-publish](https://www.nature.com/nature-portfolio/editorial-policies/self-archiving-and-license-to-publish). Those licensing terms will supersede any other terms that the author or any third party may assert apply to any version of the manuscript.

For information regarding our different publishing models please see our [Transformative Journals](https://www.springernature.com/gp/open-research/transformative-journals) page. If you have any questions about costs, Open Access requirements, or our legal

22forms, please contact ASJournals@springernature.com.

[REDACTED]

[REDACTED]

Reviewer #2:

Remarks to the Author:

The authors have addressed all of my comments in detail. I have no further suggestions for the manuscript. This is exciting work and would be of broad interest to the readers of Nature Ecology & Evolution.

Reviewer #3:

Remarks to the Author:

The authors have carefully revised the manuscript to address all my concerns. Congratulations on this excellent work.

Final Decision Letter:

19th July 2022

Dear Dr Kingsley,

We are pleased to inform you that your Article entitled "Evolution of stickleback spines through independent *cis*-regulatory changes at *HOXDDB*", has now been accepted for publication in Nature Ecology & Evolution.

Over the next few weeks, your paper will be copyedited to ensure that it conforms to Nature Ecology and Evolution style. Once your paper is typeset, you will receive an email with a link to choose the appropriate publishing options for your paper and our Author Services team will be in touch regarding any additional information that may be required

23You will not receive your proofs until the publishing agreement has been received through our system

Due to the importance of these deadlines, we ask you please us know now whether you will be difficult to contact over the next month. If this is the case, we ask you provide us with the contact information (email, phone and fax) of someone who will be able to check the proofs on your behalf, and who will be available to address any last-minute problems . Once your paper has been scheduled for online publication, the Nature press office will be in touch to confirm the details.

Acceptance of your manuscript is conditional on all authors' agreement with our publication policies (see www.nature.com/authors/policies/index.html). In particular your manuscript must not be published elsewhere and there must be no announcement of the work to any media outlet until the publication date (the day on which it is uploaded onto our web site).

Please note that *Nature Ecology & Evolution* is a Transformative Journal (TJ). Authors may publish their research with us through the traditional subscription access route or make their paper immediately open access through payment of an article-processing charge (APC). Authors will not be required to make a final decision about access to their article until it has been accepted. [Find out more about Transformative Journals](https://www.springernature.com/gp/open-research/transformative-journals)

Authors may need to take specific actions to achieve [compliance](https://www.springernature.com/gp/open-research/funding/policy-compliance-faqs) with funder and institutional open access mandates. If your research is supported by a funder that requires immediate open access (e.g. according to [Plan S principles](https://www.springernature.com/gp/open-research/plan-s-compliance)) then you should select the gold OA route, and we will direct you to the compliant route where possible. For authors selecting the subscription publication route, the journal's standard licensing terms will need to be accepted, including [those licensing terms](https://www.nature.com/nature-portfolio/editorial-policies/self-archiving-and-license-to-publish) will supersede any other terms that the author or any third party may assert apply to any version of the manuscript.

An online order form for reprints of your paper is available at <https://www.nature.com/reprints/author-reprints.html>. All co-authors, authors' institutions and authors' funding agencies can order reprints using the form appropriate to their

geographical region.

We welcome the submission of potential cover material (including a short caption of around 40 words) related to your manuscript; suggestions should be sent to Nature Ecology & Evolution as electronic files (the image should be 300 dpi at 210 x 297 mm in either TIFF or JPEG format). Please note that such pictures should be selected more for their aesthetic appeal than for their scientific content, and that colour images work better than black and white or grayscale images. Please do not try to design a cover with the Nature Ecology & Evolution logo etc., and please do not submit composites of images related to your work. I am sure you will understand that we cannot make any promise as to whether any of your suggestions might be selected for the cover of the journal.

You can generate the link yourself when you receive your article DOI by entering it here: <http://authors.springernature.com/share>.

[REDACTED]

P.S. Click on the following link if you would like to recommend Nature Ecology & Evolution to your librarian <http://www.nature.com/subscriptions/recommend.html#forms>

** Visit the Springer Nature Editorial and Publishing website at http://editorial-jobs.springernature.com?utm_source=ejp_NEcoE_email&utm_medium=ejp_NEcoE_email&utm_campaign=ejp_NEcoE for more information about our career opportunities. If you have any questions please click [here](mailto:editorial.publishing.jobs@springernature.com).**